# Membrane curvature sensing and symmetry breaking of the M2 proton channel from Influenza A

**James Lincoff[1†], Cole VM Helsell[1,2†], Frank V Marcoline[1†], Andrew M Natale[1,2‡], Michael Grabe[1]***

[1]Cardiovascular Research Institute, Department of Pharmaceutical Chemistry, University of California, San Francisco, San Francisco, United States; [2]Graduate Group in Biophysics, University of California, San Francisco, San Francisco, United States

**\*For correspondence:** michael.grabe@ucsf.edu

[†]These authors contributed equally to this work

**Present address:** [‡]Altos Labs, California, United States

**Competing interest:** The authors declare that no competing interests exist.

**Abstract** The M2 proton channel aids in the exit of mature influenza viral particles from the host plasma membrane through its ability to stabilize regions of high negative Gaussian curvature (NGC) that occur at the neck of budding virions. The channels are homo-tetramers that contain a cytoplasm-facing amphipathic helix (AH) that is necessary and sufficient for NGC generation; however, constructs containing the transmembrane spanning helix, which facilitates tetramerization, exhibit enhanced curvature generation. Here, we used all-atom molecular dynamics (MD) simulations to explore the conformational dynamics of M2 channels in lipid bilayers revealing that the AH is dynamic, quickly breaking the fourfold symmetry observed in most structures. Next, we carried out MD simulations with the protein restrained in four- and twofold symmetric conformations to determine the impact on the membrane shape. While each pattern was distinct, all configurations induced pronounced curvature in the outer leaflet, while conversely, the inner leaflets showed minimal curvature and significant lipid tilt around the AHs. The MD-generated profiles at the protein–membrane interface were then extracted and used as boundary conditions in a continuum elastic membrane model to calculate the membrane-bending energy of each conformation embedded in different membrane surfaces characteristic of a budding virus. The calculations show that all three M2 conformations are stabilized in inward-budding, concave spherical caps and destabilized in outward-budding, convex spherical caps, the latter reminiscent of a budding virus. One of the C2-broken symmetry conformations is stabilized by 4 kT in NGC surfaces with the minimum energy conformation occurring at a curvature corresponding to 33 nm radii. In total, our work provides atomistic insight into the curvature sensing capabilities of M2 channels and how enrichment in the nascent viral particle depends on protein shape and membrane geometry.

## Editor's evaluation

In this important study, the authors combined atomistic simulations and continuum mechanics models to probe how structural features of the M2 channel impact the local membrane properties and stability of the channel in membranes of different curvatures. The level of evidence provided by the computational analysis is convincing, and the insights can potentially lead to novel strategies that screen for drug molecules that stabilize fission-incompetent conformations of the M2 channel. The multi-scale approach will find utility to many problems in membrane reshaping.

## Introduction

Influenza A M2 is a small, α-helical homo-tetrameric membrane protein essential for viral replication (*Rossman and Lamb, 2011*). M2 has two primary functions in the viral life cycle. First, during cell entry via endocytosis, endosomal acidification initiates fusion of the endosomal and viral membranes, but it also activates M2 channels causing further acidification that releases bound viral ribonucleoproteins (vRNP) to enter the host's cytoplasm (*Pielak and Chou, 2011*). Second, during capsid exit from the cell, M2 channels residing in the host plasma membrane migrate to the budding virion where they enrich at the neck region and promote membrane scission resulting in the release of the mature membrane enveloped viral particle (*Figure 1A*). Consequently, M2-deletion mutants are replication impaired, resulting in accumulation of ~25 nm blebs on the host membrane that only rarely resolve into infectious particles (*Rossman et al., 2010*). While other influenza factors are involved in viral egress, M2 seems to play the main part in ESCRT-independent scission of nascent viral buds by promoting further constriction of the existing ~25 nm neck in order to facilitate spontaneous membrane scission, and thus determines the shape of the mature virion (i.e., spherical or filamentous) (*Martyna et al., 2017*; *Rossman et al., 2012*).

The M2 protein is 97 residues long with largely disordered N- and C-termini connected by a single pass transmembrane (TM) α-helical segment (residues 23–46) and an amphipathic helix (AH; residues 47–62) called the AH domain. Channels are homo-tetrameric assemblies with the ion conduction pathway formed at the center of the four TM helices, and a minimal-construct 'conductance domain' (CD) comprised of the TM and AH domains is both ion channel competent and sufficient for budding (*Sharma et al., 2010*). The cytoplasm-facing AH domain is essential for scission activity (*Roberts et al., 2013*), but M2 chimeras with genetically unrelated amphipathic helices are scission-competent and promote viral replication in vivo (*Hu et al., 2020*). Meanwhile, the N- and C-termini are not essential for bud localization, negative Gaussian curvature (NGC) generation, or scission (*Chen et al., 2008*; *Kwon and Hong, 2016*; *Martyna et al., 2017*).

High-resolution structural insight comes from X-ray and solid-state Nuclear Magnetic Resonance (ssNMR) structures solved in a variety of conditions in the presence and absence of drugs, some of which are represented in *Figure 1B*. All structures reveal channels composed of four α-helical TM segments, with varying degrees of TM helix tilt with respect to the central axis. Generally, the helices exhibit greater splay on the cytoplasmic/intraviral side of the membrane than on the extracellular/extraviral surface giving rise to a conically shaped membrane spanning region. Helix kinking at glycine 34 has been suggested to control the degree of splay in active (greater) and inactive (lesser) conformations in a pH-dependent manner, which consequently increases and decreases, respectively, the conical character of the protein (*Stouffer et al., 2008*). Unfortunately, crystal structure constructs lack the entire AH domain and hence provide limited use in understanding the full membrane deforming function of the protein *Schmidt et al., 2013*; however, several structures (mostly ssNMR) have resolved this portion revealing that it is the most variable part of the CD (*Figure 1B*). In some structures, it lies parallel to the membrane at the edge of the protein further exaggerating the conical shape of the channel (*Andreas et al., 2015*; *Sharma et al., 2010*), but in others the elbow between the TM and AH domains is extended and the AH domains become more aligned with the membrane normal. In the extreme case, the partial AH domain of 6OUG forms a continuous helix with the TM domain. Most studies reveal that the backbone adopts a roughly fourfold symmetric channel; however, an ssNMR structure of a seasonal variant of M2 provided spectroscopic evidence for a twofold symmetric tetramer (*Andreas et al., 2015*) in which the AH domains break fourfold symmetry (*Figure 1B*). Moreover, double electron–electron resonance (DEER) experiments support the notion that the AH domains are dynamic with a high degree of flexibility (*Herneisen et al., 2017*; *Kim et al., 2015*). As stated, while M2 channel activity correlates with the degree of helix splay, we focus in this manuscript exclusively on the shape of the M2 protein and not its channel activity. Despite our focus, we continue to refer to M2 as a channel, at times.

Proteins induce membrane deformation via many mechanisms (*Argudo et al., 2016*; *Farsad and De Camilli, 2003*; *Jarsch et al., 2016*; *McMahon and Boucrot, 2015*; *Nepal et al., 2018*; *Phillips et al., 2009*) of which a subset have been proposed for M2 localization and curvature generation. Proteins that bind one membrane leaflet or TM proteins with asymmetry across the bilayer midplane can impart their spontaneous curvature to the membrane via a *wedge insertion* mechanism, which has been proposed for a variety of membrane-bending proteins, such as epsins and BAR proteins.

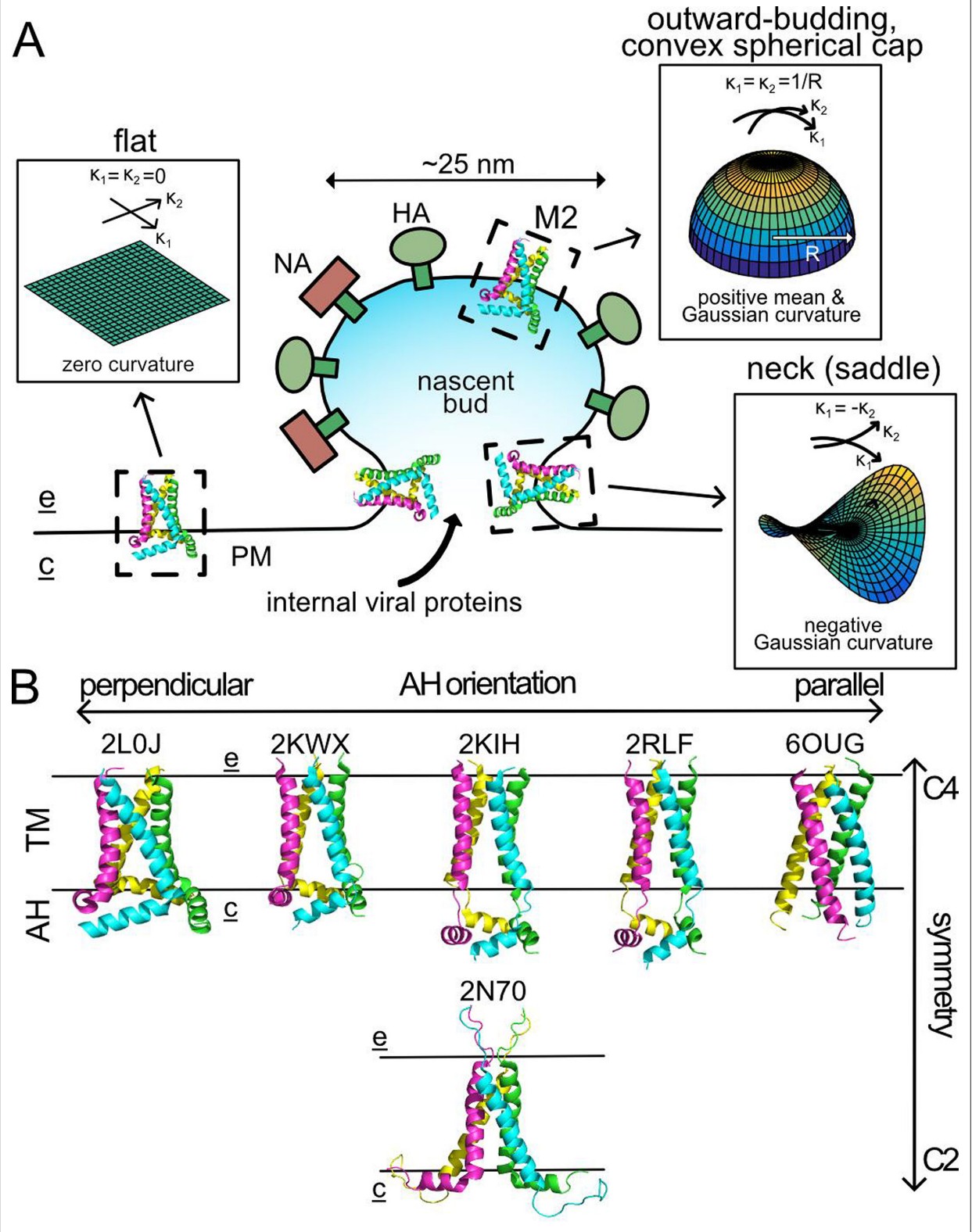

**Figure 1.** M2 channels and influenza egress. (**A**) Structural organization of the nascent viral bud. M2 first accumulates in the host plasma membrane (PM), which is approximately flat and therefore has zero Gaussian curvature. M2 then diffuses to the neck of a nascent viral bud, which has negative Gaussian curvature characterized by principal curvatures, $\kappa_1$ and $\kappa_2$, of opposite sign. Finally, a small minority of the initial M2 population migrates into the outward-budding, convex cap of the mature virion, which is otherwise enriched in viral spike proteins neuraminidase (NA) and hemagglutinin (HA). Labels e̲ and c̲ highlight the e̲xtracellular face of the PM and the c̲ytosolic face. (**B**) A sample of published M2 structures (indicated by PDB accession

*Figure 1 continued on next page*

*Figure 1 continued*

codes) grouped by reported symmetry (top row: C4 symmetry vs. bottom row: C2 symmetry) and relative position of their amphipathic helix (AH) domains: 2L0J on the far left turns sharply and lies perpendicular to the transmembrane (TM) domain, with the AH domain becoming less associated with the TM domain until 6OUG on the far right shows the AH continuing parallel to the TM helices.

This mechanism is also relevant for M2 as the conically shaped CD construct is wedge-like, displacing more membrane area on the cytoplasmic leaflet than the extracellular leaflet, imparting a spontaneous curvature on the membrane that agrees with the estimated curvatures at the budding neck (*Schmidt et al., 2013*). An additional curvature-generating role for the AH domain is suggested by the *packing defect stabilization model* which states that bulky hydrophobic residues from amphipathic helices intercalate into the membrane at curvature-deformed sites to minimize exposure of the hydrophobic tails to solution (*Cui et al., 2011*) as seen in membrane-bound ESCRT-III helical reconstructions (*Moss et al., 2023*). Finally, the nascent bud is cholesterol and sphingomyelin rich (*Gerl et al., 2012*; *Ivanova et al., 2015*), giving rise to a line tension at the domain boundary with the host membrane. M2 and other membrane scission proteins have been shown to be '*linactants*' that modify line tension (*Kuzmin et al., 2005*), supporting a role for M2 clustering at the boundary and aiding with demixing of the two phases (*Madsen et al., 2018*). A recent simulation study of M2 crowded into a bilayer patch showed the emergence of linear clusters and attendant changes in membrane curvature (*Paulino et al., 2019*). This *linactant model* provides the added benefit of explaining how influenza-specific membrane components are enriched from the host membrane (*Gerl et al., 2012*). A molecular mechanism supporting M2's role as a linactant comes from experiments showing that M2 binds cholesterol sub-stoichiometrically with two proximal (as opposed to diagonal) lipids per M2 tetramer (*Elkins et al., 2017*). This asymmetry allows M2 to satisfy its preferred cholesterol binding orientation when it localizes to the interface and one side embeds in the cholesterol-rich bud and the other faces the host membrane.

While these theories of curvature generation suggest that M2 may deform membranes, they do not address the observation that M2 migrates to the neck of budding vesicles as opposed to the spherical cap, for instance. In fact, the dominant view of M2 as a fourfold symmetric, wedge-like shape is at odds with NGC localization, because any two orthogonal axes of the protein induce spontaneous curvatures of the *same* sign (*Figure 1A*). However, the neck is a catenoid or saddle-like membrane structure characterized by two, orthogonal principal curvatures, $\kappa_1$ and $\kappa_2$, of *opposite* sign (*Figure 1A*). Moving along either principal direction, the membrane in the neck bends away from a point on the surface in different directions, and hence the Gaussian curvature, defined as $K = \kappa_1 \cdot \kappa_2$, is negative. Thus, while the shape of M2 may curvature-match the neck in one direction, it would appear to frustrate the local shape in the orthogonal direction suggesting that, if anything, the channel should be excluded from the neck. Recent coarse-grained simulations by Voth and coworkers using a fourfold symmetric model of M2 support this notion, as they found that M2 migrated to patches of positive Gaussian curvature rather than the expected NGC regions (*Madsen et al., 2018*).

One solution to this apparent contradiction is that M2 and the AHs break fourfold symmetry, as discussed above, and adopt configurations compatible with the two distinct principal curvatures present in the neck. Tangential support for such a mechanism comes from work by the Antonny lab showing that geometric rearrangements of defect-sensing helices imparted by changes in the flanking protein scaffold have modified curvature specificity in the cell (*Doucet et al., 2015*). Here, we initiated unrestrained, all-atom molecular dynamics (MD) simulations of the M2 channel and identified large conformational changes in the AH domain that broke fourfold symmetry. Inspired by these broken symmetry conformations, we next asked how the surrounding lipid bilayer responded to M2 channels with different symmetry configurations. To do this, we performed restrained simulations on three different systems: a fourfold symmetric NMR structure, a twofold symmetric NMR structure, and a twofold symmetric parallel AH domain model inspired by our unrestrained simulations. Each protein configuration induced specific deformations in the surrounding bilayer with twofold distortion patterns coming from the latter two systems and a largely radially symmetric distortion pattern emitted from the fourfold channel structure. However, all systems shared several common features, notably pronounced curvature generation in the extracellular leaflet and strong lipid tilt around the amphipathic helices in the intracellular leaflet. After noting an imbalance in the leaflet packing in our original simulations, we ran additional simulations to normalize lipid densities, highlighting the need

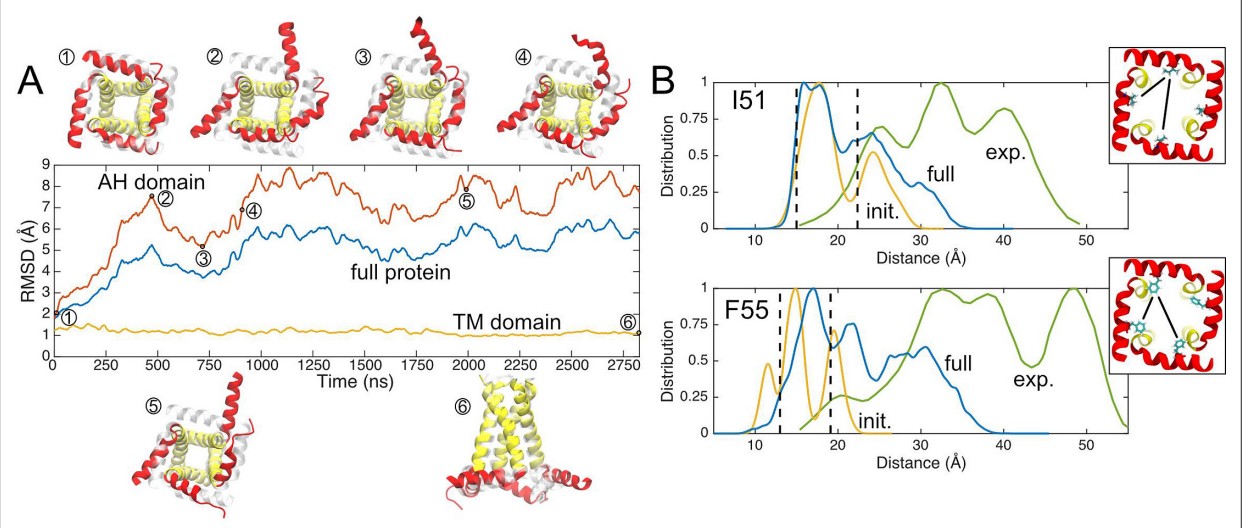

**Figure 2.** Unrestrained simulations reveal dynamic amphipathic helix (AH) domains. (**A**) RMSDs of Ca backbones over time. Yellow indicates the transmembrane (TM) domain only, red the AH domain only, and blue the full protein. Snapshots from unrestrained molecular dynamics (MD) are numbered along the *X*-axis with the starting structure overlaid in light gray, showing AH dynamics. (**B**) Double electron–electron resonance (DEER) data provided by the Howard lab (green) compared to distance estimates from simulation between labeled AH domain residues (blue), including the first 5 ns alone (yellow). Insets in upper and lower panels show the I51 (top) and F55 (bottom) residue on 2L0J (view from cytoplasm), with adjacent and diagonal distances represented by two black lines.

The online version of this article includes the following source data for figure 2:

**Source data 1.** Raw double electron–electron resonance (DEER) data from Figure 7 of *Kim et al., 2015* (green curves in panel B).

for careful membrane construction for highly asymmetric proteins. While there are quantitative differences in lipid tilt and leaflet thickness in response to repacking, the protein-induced membrane deformations remained qualitatively similar. Using the MD-derived deformations at the channel boundary as boundary conditions, we determined the membrane energetic cost associated with placing these three different protein configurations into spherical caps, flat membranes, and saddle regions. None of these configurations are predicted to be stabilized in outward-budding, convex spherical caps, while they all favor inward-budding, concave spherical caps. The fourfold symmetric 2L0J structure is not stabilized in NGCs, while one of the twofold symmetric constructs is stabilized over a range of saddle curvatures. The minimum energy of 4 kT compared to the flat bilayer occurs at a radius of curvature of ~30 nm, very close in size to the stalled particle size observed in cells infected by M2 defect mutant strains (*Rossman et al., 2010*), and it is predicted to become more energetically favorable as cholesterol content in the bud increases. Thus, our results provide a mechanical mechanism for why M2 channels are absent in the spherical cap of a nascent capsid and suggest that symmetry breaking by the amphipathic helices could explain channel localization to the neck region.

## Results

### Unbiased MD simulations reveal flexible amphipathic helical regions

To explore the conformational landscape of the M2 protein, we initiated an unbiased all-atom MD simulation of the fourfold structure (PDBID: 2L0J) in a heterogeneous bilayer consisting of 1-palmitoyl-2-oleoyl-sn-*glycero*-3-phosphocholine (POPC), 1-palmitoyl-2-oleoyl-sn-*glycero*-3-phosphatidylglycerol (POPG), and cholesterol. Specifically, the backbone of 2L0J exhibits fourfold symmetry providing the impression that it is globally symmetric, yet the individual side chains on each monomer lack any symmetry, while several histidines in the central pore (not in contact with the membrane) were built with twofold symmetry. Just after equilibration at the start of production, the channel (AH – red/TM – yellow) was quite similar to the starting NMR structure (transparent white) (snapshot 1, *Figure 2A*). However, over the first 500 ns, the total protein root mean squared deviation (RMSD) (blue curve) increased, and the AH domains (red curve) mirrored this increase but were even larger by 1–2 Å

reaching just over 7 Å at 500 ns (snapshot 2). Several AH helices have broken symmetry and moved radially away from the central axis losing contact with the other subunits, with the top helix the most pronounced in this example. In contrast, the TM domain (yellow) remained stable throughout the 2750 ns simulation with an RMSD ~1 Å. Additional snapshots along the trajectory (3–5) reinforce the notion that the AH helices are mobile, showing that they all deviate from the starting structure, some quite significantly. The helices primarily retain their secondary structure, but some partially unfold and sometimes refold (top helix in snapshot 4 vs. 5). The sixth and final snapshot shows a side view of the channel highlighting that the TM domains (yellow) retained fourfold symmetry with a very close match to the 2L0J structure.

We computed the residue-to-residue distance distribution for two AH residues (I51 and F55), which had previously been measured using DEER for the same construct in the same lipid composition (*Kim et al., 2015*). Each residue has essentially two distal heavy atom distance values in the fourfold starting structure, a smaller adjacent subunit-to-subunit value (~13 Å) and a larger diagonal value (18 or 22 Å) indicated by vertical dashed lines (*Figure 2B*). The experimental distances (green curves) are much broader and occur over larger distances ranging from 20 to 45 Å for the I51 position and 20 to 50 Å for the F55 position. Part of this difference arises from the increased length of the methanethiosulfonate spin label (MTSL) probes compared to the isoleucine and phenylalanine side chain surrogates measured from the structure, but this difference alone cannot explain the large discrepancy. The dynamic nature of the AH helices provides a picture much closer to the experimental values, as proposed by the Howard lab (*Kim et al., 2015*). We computed the inter-subunit amino acid-to-amino acid distance distributions for both residues (6 unique distances for each position in the 4-subunit channel) from the trajectory over very short times 5 ns (yellow curve) and the full 2750 ns simulation (blue curve). Initially, the side chains explore a range of rotamer conformations, despite little backbone motion in the AH domain, and this results in distributions that are several Ångströms wider than the bounds extracted from the NMR structure, but fail to account for the DEER data. As the simulations progress, the deviations in the AH domains and breaking of fourfold symmetry create a much broader range of distance values (blue curves) 15–20 Å greater than the largest values extracted from the static structure that begin to approach the experimental range. The lack of agreement between simulations and experiment is not unexpected as we are not modeling the probes explicitly, and our simulations are not converged. This latter point is evident as we see excursions in some AH helices that are not sampled in others despite the symmetry.

Our unbiased simulations confirm that the AH domains are flexible and can adopt many conformations, some of which break the fourfold symmetry of the channel. Both of these observations are consistent with the range of conformations observed in structural databases (see *Figure 1*) and with DEER experiments (*Herneisen et al., 2017*; *Kim et al., 2015*). The simulations also suggest that reorientation of the AH domains in a flat bilayer requires little energy and may be favorable, as it occurs spontaneously on a short timescale (microseconds) and in some cases helices return to their starting position (right AH helix in snapshot 2 vs. 3, *Figure 2A*).

## Protein footprint asymmetry can lead to differential leaflet stresses

Motivated by the AH domain flexibility observed in the unrestrained simulation, we wanted to understand how distinct M2 conformations influence the surrounding membrane. To efficiently sample these membrane shapes, we initiated a series of protein-restrained simulations using the same membrane building protocol employed in the unrestrained simulation. During review, however, it became clear that the leaflets exhibited differences arising from improper lipid packing during the initial build. This issue has been noted in other systems with strong asymmetry, such as those with heterogeneous leaflet compositions or asymmetric protein inclusions (*Doktorova and Weinstein, 2018*). Membranes can resolve leaflet asymmetries via lipid flip-flop or curvature generation; however, they cannot be resolved in low-microsecond MD simulations with periodic boundary conditions, giving rise to internal stresses that can alter membrane mechanical properties (*Hossein and Deserno, 2020*).

To assess the impact of asymmetry in our simulations, we evaluated the differential tension in our restrained simulation of the fourfold symmetric 2L0J structure embedded in the same membrane as the unrestrained simulation (*Figure 2*), with 200 lipids in the upper leaflet and 150 in the lower leaflet. We extended the run 100 ns with a high save rate on all positions and velocities to compute the complete three-dimensional pressure tensor throughout the simulation box using the GROMACS-LS

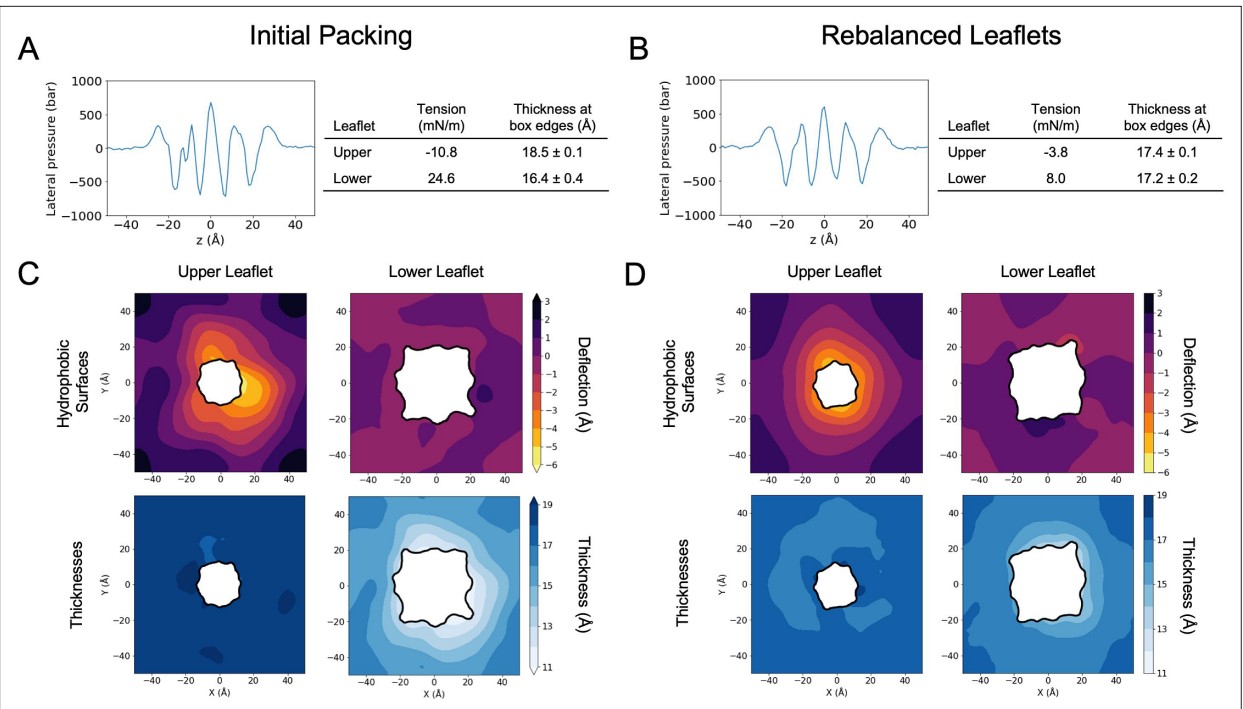

**Figure 3.** Influence of lipid packing on bilayer tension and shape. Comparison of tension and structural features of simulated membranes with the initial packing (left, panels **A, C**) versus the rebalanced leaflets (right, panels **B, D**). (**A**) Lateral pressure profile of the initial restrained 2L0J simulation, with bilayer midplane centered at z = 0. Upper leaflet in z > 0 range; lower leaflet in z < 0 range. Leaflet tensions and thicknesses at box edge (defined by calculating the average thickness excluding a 60 Å × 60 Å square cutout centered around the protein) at right. (**B**) Same as panel A for the rebalanced restrained 2L0J simulation. (**C**) Hydrophobic surfaces (top row) and leaflet thicknesses (bottom row) for the initial simulation. White cutout areas with black edges represent the footprint of the protein from the last frame of the simulation. (**D**) Same as panel C for the rebalanced simulation. Note that protein footprints in C and D differ due to motion of protein side chains.

protein (*Ollila et al., 2009*; *Vanegas et al., 2014*). From the pressure tensor, we calculated the lateral pressure profile and leaflet tensions arriving at a value of −10.8 mN/m in the upper leaflet and 24.6 mN/m in the lower leaflet (*Figure 3A*). This indicates that the upper leaflet is under compression resulting from overpacking relative to the lower leaflet, which consequently is in tension. The tension asymmetry is also evident from comparing the upper leaflet (+10 Å) and lower leaflet (−10 Å) peaks in the pressure profile, which are markedly different. This level of differential tension produces differences in the leaflet structure, which can be seen from plotting the time-averaged upper and lower hydrophobic surfaces and the bilayer thickness (*Figure 3C*). X and Y axes correspond to the size of one periodic image of the simulation box, and the inner black curves represent the shape of the membrane–protein contact curve in the upper and lower leaflets. The footprint of each channel is much larger in the lower leaflet than the upper leaflet, resulting from the presence of the AH domains and providing M2 its characteristic wedge shape. While bilayer asymmetry is expected near M2, it should vanish farther from the protein, but in this case the upper leaflet is 2 Å thicker than the lower leaflet at the boundary (*Figure 3A, C*).

We therefore conducted a series of additional restrained 2L0J simulations, keeping the number of lipids in the upper leaflet fixed while increasing the number in the lower leaflet. We evaluated each repacked simulation according to its leaflet tensions and structural properties including the area per lipid, thickness, and tilt (see Appendix: Section 1 for the full dataset). We eventually arrived at a rebalanced condition, with 183 lipids in the lower leaflet and 200 in the upper leaflet, that restores symmetry to the leaflet tensions as can be seen by comparing the pressure profile peaks at z = ±10 Å (*Figure 3B*). Additionally, the upper and lower leaflet thicknesses at the box edges match within error (<0.2 Å difference), indicating that the lipid densities are balanced (*Figure 3B*). Interestingly, the shapes of the upper and lower leaflet surfaces are insensitive to repacking. In both cases, the lower leaflet surface remains flat, while the original upper leaflet surface exhibits a subtle, sub-Ångström greater deflection at the protein boundary (*Figure 3C, D*) [Note that throughout we use the term

deflection from structural mechanics to refer to membrane deformationsthat result in displacements along the Z-axis, normal to the plane of the membrane.]. The membrane shape is one of the most robust features of this system, as changes were minor across the wide range of lower lipid counts explored despite dramatic changes in other membrane properties, such as tension and lipid shape (see *Appendix 1—table 1*). There is some remaining differential tension, −3.8 and 8.0 mN/m in the upper (compression) and lower (tension) leaflets, respectively, which likely arises from the curvature imposed on the membrane by the protein coupled with the periodic boundary conditions. Nonetheless, the symmetry in area per lipid, leaflet thicknesses, and lipid tilt values at the periodic box edges (see Appendix: Section 1)—indicates that the remaining tension does not unduly affect the membrane structure (*Hossein and Deserno, 2020*).

All further simulations were conducted with the 200/183 (upper/lower) lipid ratio, but before moving forward we returned to the initial unrestrained 2L0J simulations to determine if the high tensions, likely present in these systems, caused the high AH mobility observed in *Figure 2*. We performed four independent simulations starting from repacked, low tension configurations and found that the amphipathic helices exhibited a similar high degree of conformational flexibility and symmetry breaking as before (see *Appendix 1—figure 4*).

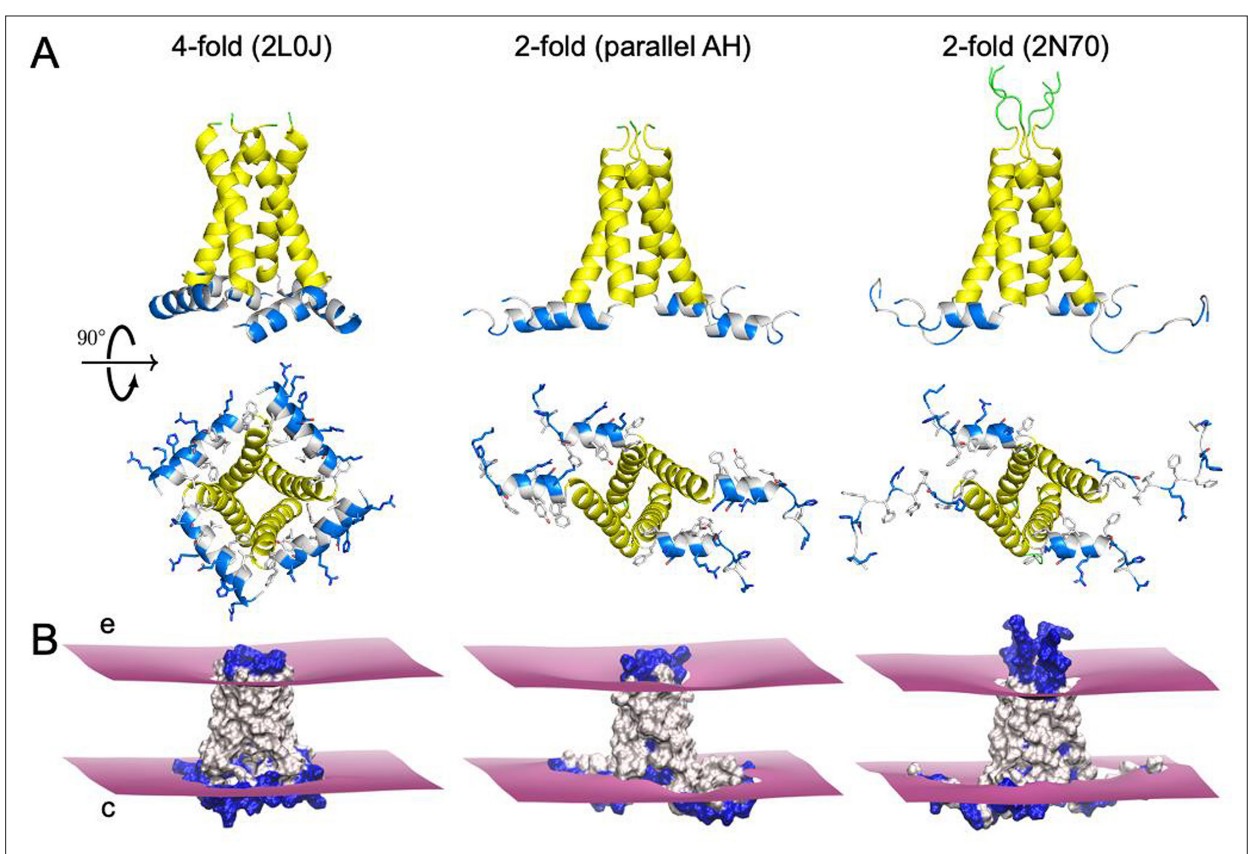

**Figure 4.** M2 membrane deformation patterns from simulation. (**A**) Structures for restrained-protein simulations. Extracellular loops in green, transmembrane (TM) domain in yellow, amphipathic helix (AH) domain in blue (polar/charged) and white (hydrophobic). Left: the fourfold solid-state NMR structure (PDBID 2L0J); center: parallel AH domain model (see main text for details); right: twofold solid-state NMR structure (PDBID 2N70). (**B**) Upper leaflet (extracellular face, **e**) and lower leaflet (cytosolic face, **c**) mean hydrophobic surfaces computed from molecular dynamics (MD; purple). Polar and charged side chains shown in blue, hydrophobic side chains in white. Structure of the parallel AH domain construct used in simulations, built from PDBID 2N70.

The online version of this article includes the following source data for figure 4:

**Source data 1.** Structure of the parallel amphipathic helix (pAH) model.

## M2 channels deform the membrane

Next, we wanted to quantitatively investigate the characteristics of the membrane around the M2 channel and explore how its properties depend on the conformation of the protein. Specifically, we were interested in the role that C4- and C2-symmetric channels play in patterning the local membrane shape, and we probed this question by carrying out protein-restrained, all-atom MD simulations on three different conformations: (1) the fourfold 2L0J structure (*Sharma et al., 2010*), (2) the twofold 2N70 structure from the Griffin lab (*Andreas et al., 2015*), and (3) a twofold symmetric parallel AH domain model based on conformational changes observed in our unbiased simulations of 2L0J (*Figure 4A*). By restraining the backbone conformations to the starting structures, we could explore how the membrane relaxed around specific channel conformations to explicitly link protein shape to membrane deformation. The parallel AH domain structure was inspired by the anecdotal observation in our unrestrained 2L0J simulation that AH domain helices from adjacent subunits break symmetry and point parallel to each other in the plane of the membrane (snapshots 3 and 5 in *Figure 2A*). We expected that such an arrangement would also break symmetry in its patterning of the membrane, potentially inducing different curvatures and hence influencing the curvature sensing of the channel. To create this model, we started from the 2N70 structure, removed the unfolded AH domains, and replaced them with one of the folded AH domains from the adjacent subunits.The two helices are largely parallel to each other, similar to the top and right helices in snapshot 5 shown in *Figure 2A*, with a slight rotation along the AH helical axis.

The organization of each channel is pictured in *Figure 4A* shown from both the side (membrane view) and cytoplasm. The cytoplasmic view highlights the distinct symmetry of each configuration, and while the parallel AH domain model and 2N70 are C2 symmetric, their overall configurations are quite distinct due in large part to the extended AH domains of 2N70. The AH domain is amphipathic consisting of charged and polar (both in blue) and hydrophobic residues (white) with the side chains explicitly represented in each structure. As expected for an AH, the charged/polar residues point down into solution and the hydrophobic residues point up into membrane core. This is also true for the amino acids on the extended AH domains of 2N70 as the phenylalanines (white) and acidic/basic (blue) residues extend their side chains in opposite directions.

The mean upper and lower leaflet surfaces, representing the interface between the lipid head-groups and tails, from our protein-restrained MD simulations (*Table 1*: simulations #8, 13, and 14) are depicted around each protein, which are shown in surface view with hydrophobic residues white and charged and polar residues blue (*Figure 4B*). These surfaces are also visualized in 2D, along with the hydrophobic bilayer thickness, in *Figure 5*. The charged portions of the AH domains fall

**Table 1.** List of simulations.

| ID | Label | Gromacs | Length (ns) | PDBID | Bilayer # lipids | Restraints |
|----|-------|---------|-------------|-------|------------------|------------|
| 1 | Unrestrained 1 | 2020.6 | 2829 | 2L0J | 200 upper; 150 lower | No |
| 2 | 2L0J 1 | 2020.6 | 3808 | 2L0J | 200 upper; 150 lower | Yes |
| 3 | pAH 1 | 2020.6 | 3760 | Parallel AH | 200 upper; 150 lower | Yes |
| 4 | 2N70 1 | 2020.6 | 1727 | 2N70 | 200 upper; 150 lower | Yes |
| 5 | 2L0J Repack 1 | 2020.6 | 2740 | 2L0J | 200 upper; 158 lower | Yes |
| 6 | 2L0J Repack 2 | 2020.6 | 2500 | 2L0J | 200 upper; 166 lower | Yes |
| 7 | 2L0J Repack 3 | 2020.6 | 2890 | 2L0J | 200 upper; 174 lower | Yes |
| 8 | 2L0J Repack 4 – Final | 2020.6 | 5000 | 2L0J | 200 upper; 183 lower | Yes |
| 9 | Unrestrained 2 | 2020.6 | 2650 | 2L0J | 200 upper; 183 lower | No |
| 10 | Unrestrained 3 | 2020.6 | 3000 | 2L0J | 200 upper; 183 lower | No |
| 11 | Unrestrained 4 | 2020.6 | 3000 | 2L0J | 200 upper; 183 lower | No |
| 12 | Unrestrained 5 | 2020.6 | 3000 | 2L0J | 200 upper; 183 lower | No |
| 13 | pAH Repack Final | 2020.6 | 3000 | Parallel AH | 200 upper; 183 lower | Yes |
| 14 | 2N70 Repack Final | 2020.6 | 3000 | 2N70 | 200 upper; 183 lower | Yes |

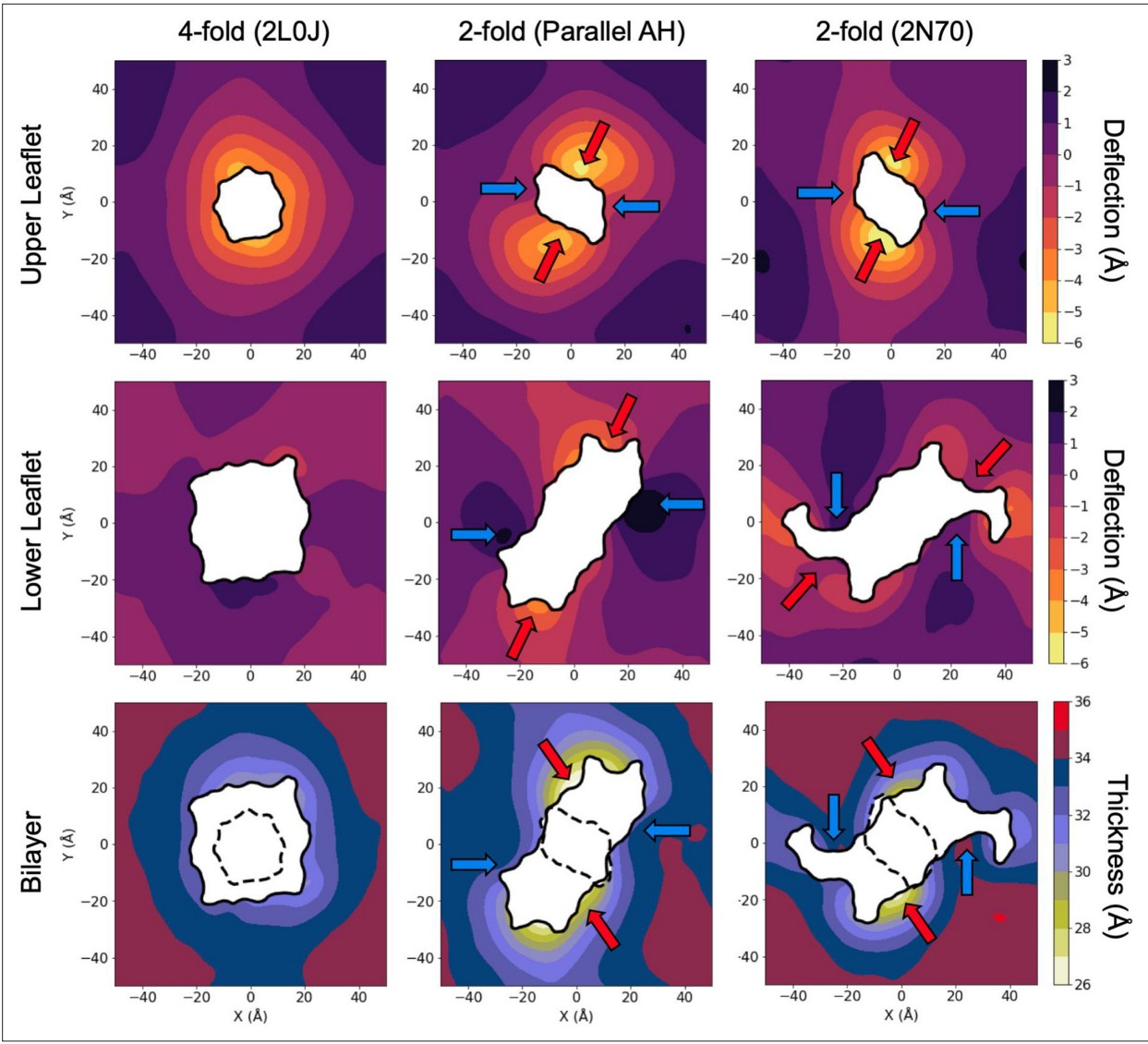

**Figure 5.** Surface height and thickness heatmaps for three different channel configurations. Compressive deflections are negative/shown in bright colors. Expansive deflections are positive/dark colors. The same scale is used for the upper and lower leaflets to highlight the greater amount of deflection in the upper as compared to the lower leaflet. Arrows correspond to regions of high (red) and low (blue) compression noted in the text.

primarily below the lower surface indicating that they are embedded in the headgroup regions or directly exposed to solution. Meanwhile, the residues exposed to the membrane core between the upper and lower leaflets are almost entirely hydrophobic. In all three cases, the membrane pinches, or compresses, as it approaches the channel, and this feature is accompanied by a pronounced curvature in the extracellular leaflet, while the cytoplasmic leaflet remains rather flat. Despite the presence of conserved general features, the deformation patterns are different.

Heatmaps of the upper (top row) and lower (middle row) membrane surfaces from *Figure 4B* provide a more detailed view of the distortions (*Figure 5*). The membrane hydrophobic surfaces are interpolated per frame to the first carbon atoms of each POPC and POPG lipid tail, then averaged over production. See Methods for further details. The inner black curves represent the shape of the membrane–protein contact curve in the upper and lower leaflets. As noted earlier, the AH domains make the channel much larger in the lower leaflet than the upper leaflet, and the four- versus twofold symmetry of each conformation is apparent in the contact curves. Membrane-thinning deflections of the surface, downward for upper leaflet and upward for the lower leaflet, are signed as negative, and

outward deflections (membrane thickening) are positive. The mean nominal undistorted height of each leaflet was set to zero in each surface based on the average height over each surface.

The membrane distortion for 2L0J (left column in *Figure 5*) shows a high degree of rotational symmetry consistent with its C4 symmetry. At the membrane–protein contact curve, the upper surface is deflected downward by −4 to −5 Å while the lower leaflet surface is largely flat, with no clear patterning from protein-associated deflection even near the amphipathic helices. Moving radially away from the membrane–protein contact curve, the upper deflection returns to its equilibrium value about 12–15 Å from the protein. Thus, the upper leaflet exhibits membrane-bending curvature, while the hydrophobic thickness of the membrane (bottom row) reveals a pronounced 2–4 Å pinch adjacent to the protein. In our simulations, far from the proteins the membrane adopts an equilibrium thickness of ~35 Å consistent with experiments suggesting phosphatidylcholine (PC):phosphatidylglycerol (PG):cholesterol bilayers are ~33 Å thick (*Tong et al., 2012*). The degree of pinching will likely vary for different lipid compositions.

Like the fourfold 2L0J system, the twofold structures induce bending and pinching in the upper leaflet. However, pinching is not rotationally uniform, as in the case of 2L0J, with both the parallel AH domain model (middle column) and the 2N70 structure (right column) producing twofold rotationally symmetric distortion patterns (*Figure 5*). For each structure, the largest thinning deflections are oriented along the *Y* axes, with less negative deflections along the *X* axes (red and blue arrows in top row, respectively). The parallel AH domain model induces a −4 to −6 Å deflection at these points of contact between the protein and membrane, but only 0 to −3 Å deflection along the *X*-axis contact points. Comparing the location of the strong upper leaflet, downward deflections with the lower leaflet protein footprints (middle row), shows that they occur above membrane embedded amphipathic helices.

As with 2L0J, the lower leaflets around both of the twofold structures show little variation in height, though the twofold nature is still present in the pattern (*Figure 5*). Similar to the upper leaflet, the lower leaflet is maximally thinned at the ends of the AH domains (red arrows), with less extreme deflections in the same locations as the upper leaflet (blue arrows). Taking the upper and lower deflections together, for the parallel AH domain model the bilayer is pinched by ~8 Å along the *Y*-axis, while showing very little compression along the *X*-axis (bottom row, middle panel). The resulting bilayer compression pattern for 2N70 is similar with symmetric regions of high compression, or pinch, along the edge of the folded amphipathic helices on the *Y*-axis and symmetric regions of low compression along the edges of the unfolded amphipathic helices along the *X*-axis (bottom row, right panel). The different channel conformations thus produce distinct local membrane deformation patterns that reflect their individual symmetries and AH orientations, suggesting that such conformational changes may be key to M2 enrichment in the distinct membrane geometries seen in experiment (*Rossman et al., 2010*).

We gain further insight into the membrane structure by calculating the average location of the bilayer midplane surface (*Figure 6*, bottom row) from which we determine the thickness of each leaflet (*Figure 6*, top and middle rows). The bilayer midplane is calculated in a similar method to the hydrophobic surfaces (*Figure 5*), based on interpolation of the last carbon atoms of each POPC and POPG lipid tail per simulation frame. See Methods for further details. While the upper leaflet surface height is strongly deformed around 2L0J, its thickness is nearly uniform (top left) because the midplane surface height (lower left) mirrors the upper leaflet surface deflection (*Figure 5*, upper right). This result has implications for the lipid tilt in the upper leaflet that we return to later. There is unevenly sampled thinning of 2–3 Å around the protein footprint in the lower leaflet (middle left), which drives the thinning of the bilayer observed in *Figure 5*.

The twofold symmetric structures produce twofold rotationally symmetric patterning in leaflet thicknesses (middle and right columns, *Figure 6*), with regions of greater thinning (red arrows) and lesser thinning (blue lines) occurring at locations where the surfaces show larger and smaller deflections in the height, respectively (*Figure 5*). Thus, unlike 2L0J, the lower membrane thickness adjacent to the protein varies greatly. The midplane heights in each of the twofold cases also exhibit extreme variation with pronounced downward deflections separated by 180° above each pair of amphipathic helices (red regions within dashed lines). These depressions arise because the AHs exclude lipid headgroups, requiring the tails to wrap around the helices to reach the TM segments. Wrapping reduces the vertical extent of the tails along the membrane normal causing the midplane depression.

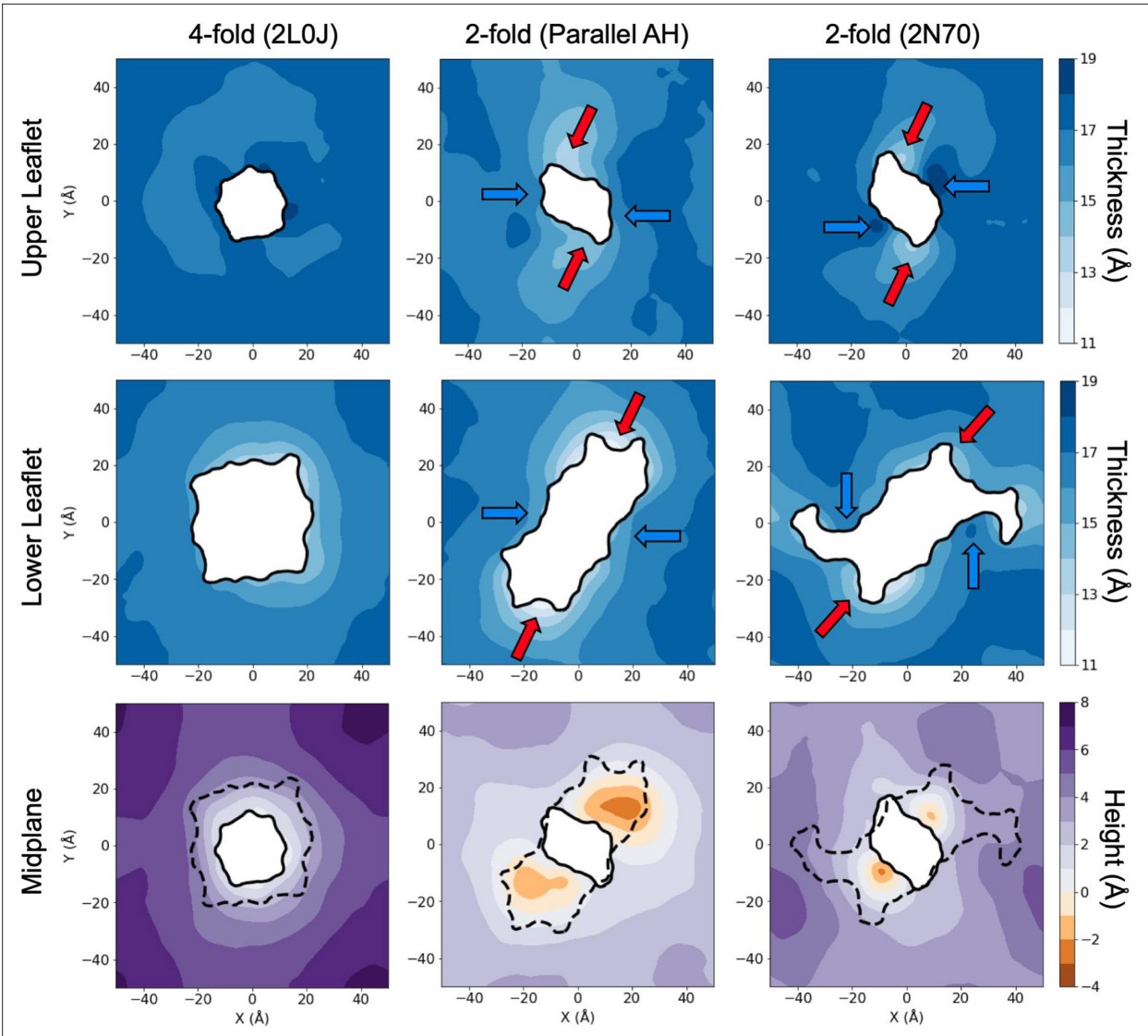

**Figure 6.** Leaflet thicknesses and position of the bilayer midplane. Upper and lower leaflet thicknesses are plotted as time-averaged surface values in the top and middle rows, with lighter shades of blue denoting regions of leaflet thinning. The height of the bilayer midplane is plotted in the bottom row. For each system, the midplane height is calculated in reference to the average position of the four L36 residues of the tetramer. These residues sit at exactly the midplane height ($z = 0$) in the 2L0J simulation. Deflections above this height are shown in purple, deflections below are shown in orange.

Accordingly, these lower leaflet lipids are deformed and tilted. The lower leaflet is undeformed, at its equilibrium thickness, at points where it directly contacts the vertical TM helices of the protein (blue arrows along the *X*-axis contact points, where the upper and lower leaflet protein footprints overlap), and these regions have thickness values close to the equilibrium value. The 2N70 structure exhibits less midplane deflection and reduced lower leaflet thinning (blue arrows) compared to the parallel AH model due to its partially unfolded helices that penetrate the membrane to a lesser degree (*Figure 4A, C*).

## Lipids tilt around the amphipathic helices

We first qualitatively assessed the equilibrated lipid structure around isolated snapshots of the proteins (*Figure 7*, top row). The upper leaflet curvature discussed previously and shown in *Figure 5* is evident in each case with the lipids deflecting downward as they approach the protein, while the lower leaflet surfaces are all comparatively flat. Lipid tilt is apparent throughout; however, it occurs to a greater degree in the lower leaflet, arising from a mixture of lipid tail kinking (e.g., the purple POPC in the

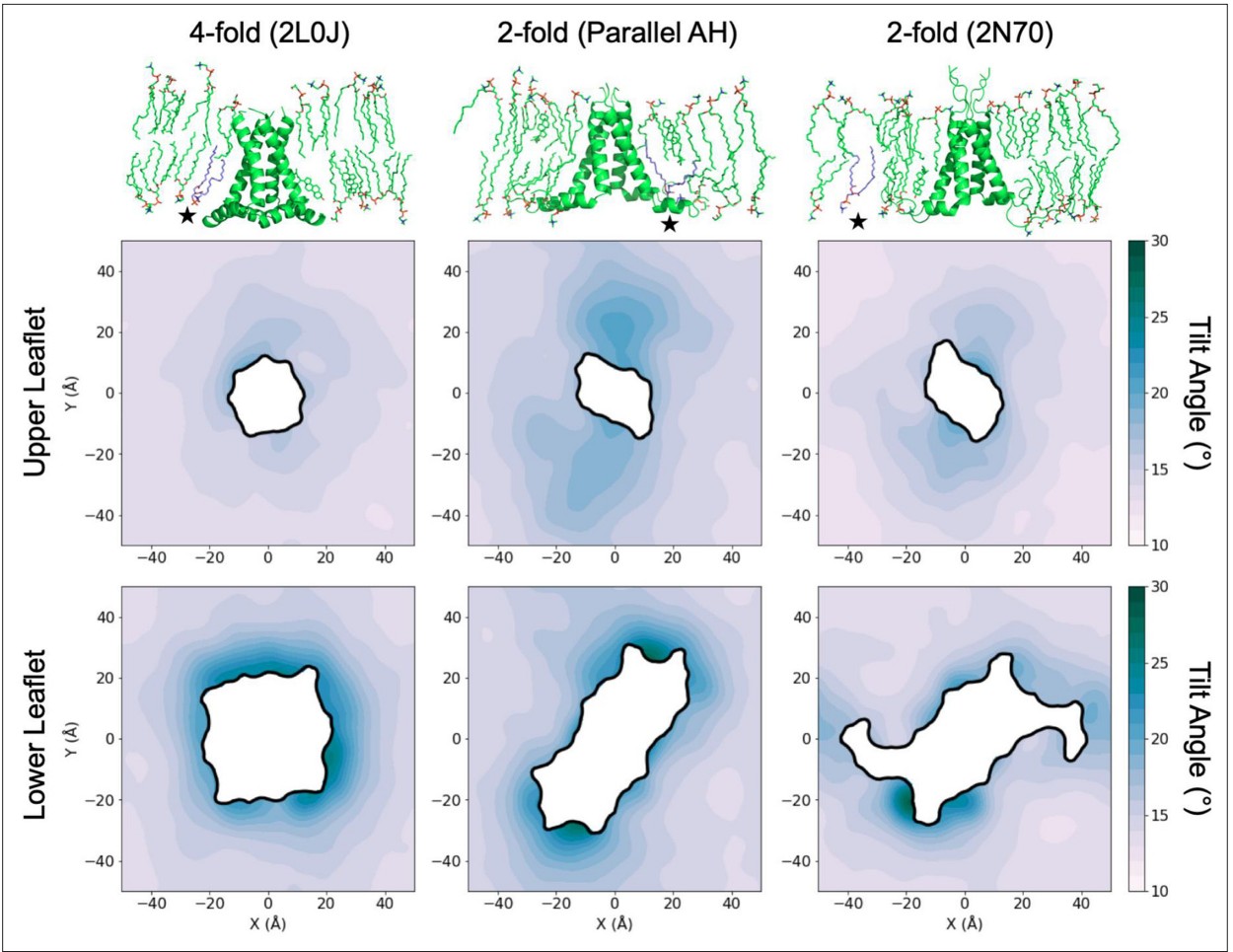

**Figure 7.** Lipid tilt around different M2 conformations. The top row shows representative all-atom snapshots extracted from the protein-restrained, equilibrium simulations of 2L0J, parallel amphipathic helix (AH) domain model, and 2N70 (2, 3, and 4 in *Table 1*, respectively). Black stars in each snapshot highlight purple lipids discussed in the main text. Two-dimensional tilt surfaces computed for each simulation in the top row separated out by upper (middle row) and lower (bottom row) leaflets. The mean lipid tilt at a given position in the *X–Y* plane with respect to the *Z*-axis (membrane normal) is reported in degrees and color coded according to the scale on the right. Cholesterol is excluded from the tilt calculations. Molecular images of the proteins are oriented as in *Figure 4*, with the viewpoint origin in the lower right quadrant of the tilt surfaces shown in the middle and bottom rows. Full size images are included as source data.

The online version of this article includes the following source data for figure 7:

**Source data 1.** Full size image of lipids around 2L0J.

**Source data 2.** Full size image of lipids around the parallel amphipathic helix (AH) domain model.

**Source data 3.** Full size image of lipids around 2N70.

2N70 snapshot), splaying apart of lipid tails (e.g., the purple POPC in the parallel AH domain model snapshot), and simple off-normal tilting of an otherwise 'regular conformation' lipid (e.g., the purple POPG in the 2L0J snapshot). These various tilt behaviors are more pronounced near the proteins, as lipids reorient to conform to the irregular wedge shape of the channel created by the partial insertion of the AH domains in the lower leaflet. Small amounts of tilt can be seen in the upper leaflet for those lipids that abut the tilted TM helices, though they retain conformations much more typical of the bulk lipids farthest from the protein. Of note, while we observe cholesterol adjacent to the channel, and cholesterol has been shown to bind M2 (*Elkins et al., 2017*), our simulations failed to identify any binding events or hot spots on the low-microsecond timescale, which may be the result of the backbone restraints or inadequate sampling.

We next characterized the lipid tilt in the three restrained simulations. Mean tilt values of all POPC and POPG lipids (excluding cholesterol) were then estimated by calculating the angle between the

Z-axis, which we used as a surrogate for the membrane normal, and the vector pointing from the head-group phosphate to the mean position of the last two carbon atoms in each tail. For all three systems, the mean tilt far from the protein approaches similar values in the upper and lower leaflets, indicating that lipids near the boundary sample equilibrium, flat bilayer-like conformations (*Figure 7*, middle and bottom rows). The patterning of the upper leaflet tilt surfaces corresponds with the patterning in the surface thickness (*Figure 6*), matching the expectation that increased lipid tilt shortens the length of the lipids along the Z-axis leading to membrane thinning. The highest degree of tilt occurs at the protein–membrane contact curve, and while there is some tilt in the upper leaflet most notably for the twofold parallel AH model (middle column) and 2N70 (right column) structures, the lower leaflets (bottom row) exhibit much more tilt around 30° at the protein and decaying to 15° at the simulation boundary. This observation is consistent with hypotheses regarding the structure of lipids adjacent to partially inserted amphipathic helices put forward by May and co-workers (*Zemel et al., 2008*), reviewed by *Zimmerberg and Kozlov, 2006*, and recently observed with brominated lipid probes and coarse-grained MD in an ESCRT-III system (*Moss et al., 2023*). The tilt pattern in the lower leaflet for the fourfold 2L0J shows a high degree of rotational symmetry; however, the twofold symmetric constructs produce less-rotationally symmetric tilt patterns. Furthermore, there are two points along

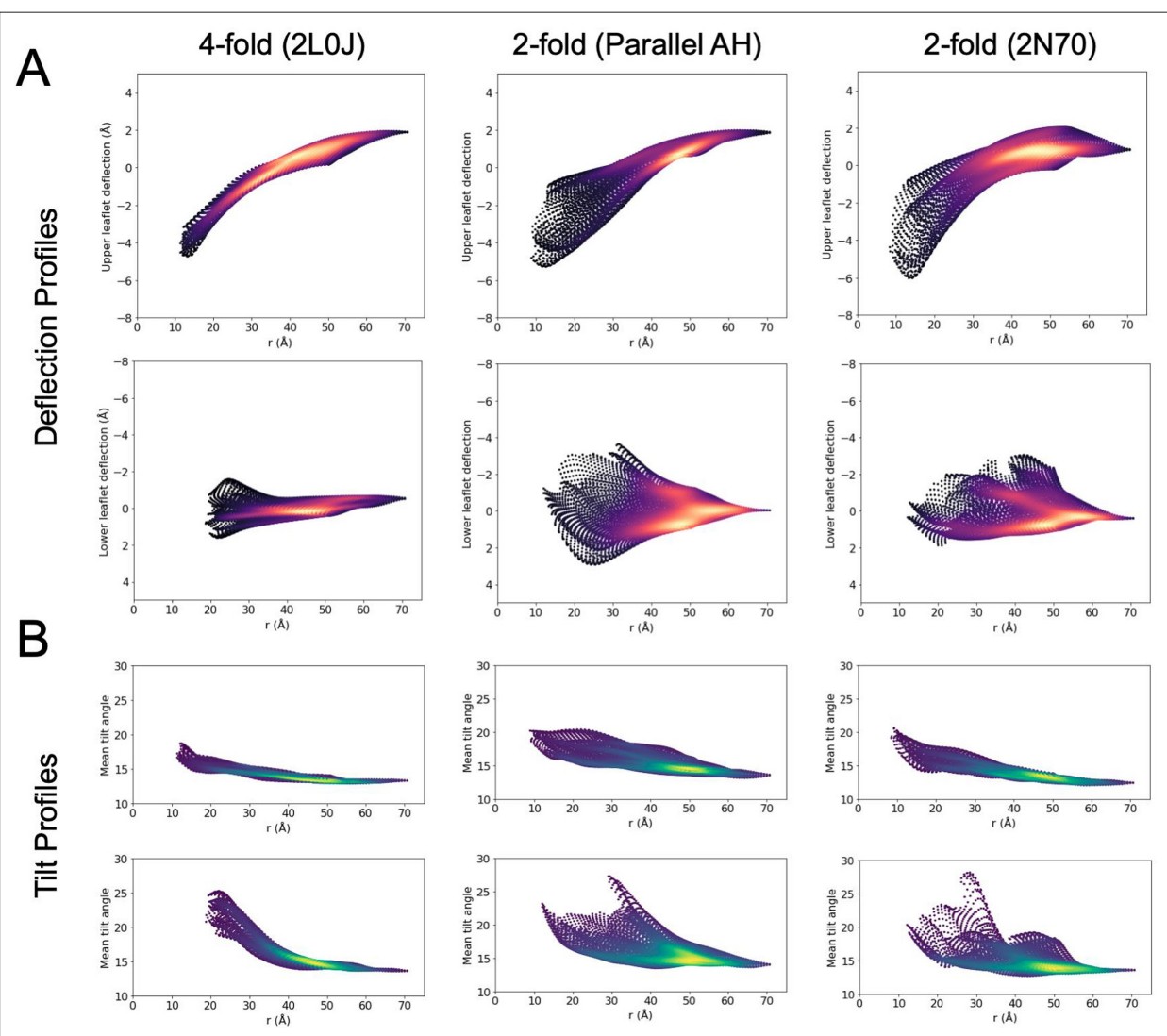

**Figure 8.** Mean deflection and tilt as a function of distance from the protein center. Mean deflection (**A**) and tilt (**B**) as a function of radial distance from the protein center for restrained simulations. Points are colored by local density, such that bright spots reflect highly sampled values of deflection/tilt versus distance. At a fixed value of distance, a vertical scan highlights the range of mean deflections/tilts sampled and the nature of the distribution of mean deflections/tilts seen at that distance.

the lower leaflet footprint of 2N70 where lipids sample near-bulk levels of low tilt, in the same areas of near-normal leaflet thickness as seen for 2N70 (blue arrows in *Figure 6*; middle row right column). These points are near the unfolded portion of the two opposing helices of 2N70 that dip out of the membrane surface, allowing the lipids to sample lower levels of tilt.

To further explore the relationship between leaflet surface deflection and lipid tilt, we plotted the mean surface deflections and lipid tilt values for each leaflet as a function of distance from the protein center in *Figure 8A and B*, respectively. Each grid point is colored according to the local density, such that bright spots indicate regions (on the *X*–*Y* simulation grid) where there is a high density of points that sample similar values of mean deflection, or tilt, versus the radial distance from the box center. Scanning vertically through the swath of points at a fixed value of distance thus highlights characteristics of the distribution of tilt or deflection at that distance. The range that is sampled reflects whether the distribution is narrow or broad, and the number of brighter spots at a fixed distance reflect whether there is a single common mean deflection (or tilt) or multiple frequently sampled mean deflections (or tilts), reflecting the full rotational versus twofold symmetry in membrane patterning for 2L0J versus 2N70 and the parallel AH model.

As expected from the deflection and tilt surfaces in *Figures 5 and 7*, respectively, the 2L0J simulation (left column, *Figure 8B*) produces the most isotropic patterning versus distance of the three simulations, with smaller ranges in value of deflection and tilt at fixed distances and brighter spots consistent with tight clustering. In the upper leaflet, the tilt experiences a modest decrease from a mean value of 17° at the protein to 14° at 40 Å and beyond. The lower leaflet displays a much stronger decay of tilt, arising from the higher tilt maximum near the protein edge, with a uniform distribution of 18–25° near the protein that decreases to the same mean value of 14° observed in the upper leaflet. In contrast, the deflections in the lower leaflet are quite uniform across all distances, with a slightly higher spread at the protein–membrane contact surface due to the geometry of the amphipathic helices. Meanwhile, the upper leaflet experiences a pronounced, and radially uniform, downward deflection by 6 Å as it approaches the protein from the box edge. Examining the snapshots at the top of *Figure 7* or *Figure 8A*, at the point of contact with the protein the membrane normal does not align with the *Z*-axis. Taking this curvature into account, we surmise that the calculated tilts in both leaflets adjacent to the protein would be slightly reduced, leaving the values at the boundary unaffected. Thus, the upper leaflet likely experiences even less tilt than calculated.

The ranges of tilts seen in the twofold symmetric simulations are consistent with 2L0J, but the distributions are much broader. The twofold parallel AH domain structure (middle column) shows multiple tilt modes in the upper leaflet at a distance of 30 Å from the protein center, at 15° and 18°, highlighted by the two bright streaks. Two dominant tilt modes are also present in the lower leaflet immediately around the protein, highlighting the $X > 0$ versus $X < 0$ anisotropy seen in *Figure 5*, and they merge to a single peak around 50 Å. The 2N70 simulation (right column) has less anisotropy, with a tighter tilt distribution throughout the upper leaflet. The upper leaflet deflection profiles for the twofold systems are similar to each other, both with tight distributions at the box edge, a large downward deflection by ~6 Å as the surface approaches the protein, and then a broad distribution of deflection values at the edge of the protein, likely due to the placement of the cytoplasmic AH domains. Their lower leaflets sample broader ranges of deflection than 2L0J, with the twofold symmetry reflected by multiple peaks at $r = 35$–40 Å; however, these peaks are within ±2 Å of the mean leaflet height, and the lower leaflets remain flatter than the upper leaflets.

Contrasting the height deformation profiles in *Figure 5* with the tilt profiles in *Figure 7*, a striking feature emerges: all M2 conformations induce large curvature in the upper leaflet with far less curvature in the lower leaflet, while the lipid tilt profiles experience little deviation from a flat bilayer in the upper leaflet and strong deviation in the lower leaflet. At the atomic level, the snapshots in *Figure 7* explain how the different M2 channels confer this behavior. The lower leaflet lipids adjacent to the AH domains kink and tilt their tails to wrap around the helices. These deformations reduce the vertical extent that the tails can reach across the membrane, and the upper leaflet lipids at the protein interface extend and move down to fill any gaps, which produces the pinching observed in the outer leaflet. Meanwhile, the extracellular span of M2 presents a cylinder-like surface to the surrounding lipids compatible with small lipid tilt angles aligned along the membrane normal. These geometric features are very similar to the curvature stresses induced by antimicrobial peptides, originally proposed by the Huang lab (*Huang et al., 2004*).

**Table 2.** Default elastic membrane material properties.

| Parameters | % cholesterol | Values | Reference |
|---|---|---|---|
| Membrane thickness ($L_C$) | 0 | 28.5 Å | *Argudo et al., 2017* |
| | 30 | 35 Å | This manuscript |
| | 50 | 37 Å | Scaling from *Ferreira et al., 2013* |
| Surface tension ($\alpha$) | 0 | $3.0 \times 10^{-13}$ N/Å | *Latorraca et al., 2014* |
| | 30 | $3.0 \times 10^{-13}$ N/Å | " " |
| | 50 | $3.0 \times 10^{-13}$ N/Å | " " |
| Bending modulus ($K_C$) | 0 | 29 $k_BT$ | Average value described in Methods |
| | 30 | 65 $k_BT$ | Scaling from *Henriksen et al., 2006* |
| | 50 | 68 $k_BT$ | Scaling from *Pan et al., 2009* |
| Gaussian modulus ($K_G$) | 0 | $-26$ $k_BT$ | Relation from *Hu et al., 2012* |
| | 30 | $-56$ $k_BT$ | " " |
| | 50 | $-61$ $k_BT$ | " " |
| Areal compression modulus ($K_a$) | 0 | $2.13 \times 10^{-11}$ N/Å | *Henriksen et al., 2006* |
| | 30 | $3.55 \times 10^{-11}$ N/Å | " " |
| | 50 | $3.73 \times 10^{-11}$ N/Å | Scaling described in Methods |

## A continuum membrane model

Next, we wanted to estimate the membrane deformation energy associated with each of these three M2 configurations to determine whether they would preferentially migrate to cellular regions of specific curvature such as the flat, saddle, or spherical caps relevant to a nascent viral bud (*Figure 1A*). Unfortunately, membrane deformation energies cannot be directly determined from our atomic simulations without employing sophisticated free-energy calculations (*Fiorin et al., 2020*), and there is no atomistic framework for determining the relative energetics of these membrane proteins in different background curvature fields, as the periodic boundaries in all-atom MD are incompatible with simple saddle shapes and spherical caps. Thus, we decided to apply continuum membrane mechanics to attempt to estimate these energies. Previously, we developed a hybrid atomistic–continuum approach for determining the insertion energies and induced membrane distortions of integral proteins (*Choe et al., 2008*) that we extended to accurately represent the membrane around proteins of complex shape (*Argudo et al., 2017*). The membrane model is based on a *Helfrich, 1973* to account for elastic curvature in each leaflet coupled with a 'mattress model' accounting for compression between the leaflets (*Huang, 1986*):

$$G_{mem}^+ = \underbrace{\frac{1}{2}\int_\Omega \frac{K_C}{2}(\nabla^2 u^+)^2 dxdy}_{mean\ curvature - bending} + \underbrace{\int_\Omega \frac{K_a}{L_C^2}(u^+ - C_M)^2 dxdy}_{compression} + \underbrace{\frac{1}{2}\int_\Omega \frac{\alpha}{2}(\vec{\nabla} u^+)^2 dxdy}_{surface\ tension} + \underbrace{\int_\Omega \frac{K_G}{2}\left(\left(\frac{\partial^2 u^+}{\partial x^2} \times \frac{\partial^2 u^+}{\partial y^2}\right) - \left(\frac{\partial^2 u^+}{\partial x \partial y}\right)^2\right) dxdy}_{Gaussian\ curvature}$$

(1)

where for brevity we have only written the energy terms for the upper leaflet, $u^+$ is the deviation from a flat equilibrium height for the upper leaflet, $K_C$ is the mean bending modulus, $K_a$ is the areal compression modulus, $K_G$ is the Gaussian bending modulus, $\alpha$ is the surface tension, $C_M$ is the bilayer compression surface, and the integrals are carried out over the entire $X, Y$ extent of the membrane domain $\Omega$. The surface $C_M$ is defined as the areal compression weighted mean of the upper and lower surfaces following the work of May and colleagues (*Fosnaric et al., 2006*), and we assume $K_a$ is the same in both leaflets. For a detailed description of the model parameters, please see the Methods section and *Table 2*. Minimizing the full free energy for both leaflets, $G_{mem}$, results in a set of Euler–Lagrange equations that are solved to determine the membrane distortions and subsequently the membrane distortion energy (*Argudo et al., 2017*). These solutions require numerically solving a fourth-order boundary value problem that involve setting the displacement and slope of the membrane at the

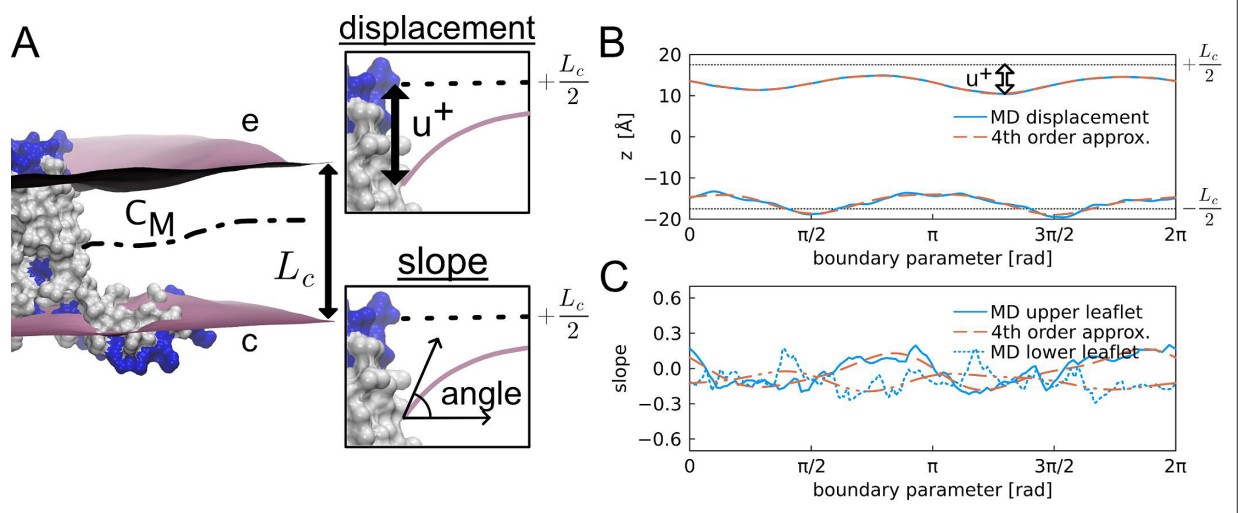

**Figure 9.** Boundary conditions extracted from molecular dynamics (MD) simulations. (**A**) Parallel amphipathic helix (AH) domain M2 protein in a lipid bilayer. The molecular surface is shown with hydrophobic residues in white and charged and polar residues in blue. The mean hydrophobic core of membrane, with equilibrium thickness $L_c$, is between the purple surfaces of the upper/extracellular leaflet e and lower/cytosolic leaflet c. The mid-plane surface between the upper and lower leaflets is $C_M$. The insets show the boundary conditions for the upper leaflet extracted at the lipid-excluded surface of the protein: (top inset) the vertical displacement from equilibrium of the upper leaflet $u^+$ and (bottom inset) the slope normal to the protein. (**B**) Membrane upper and lower bounds of the mean hydrophobic core at the protein. Solid blue: the upper and lower mean leaflet locations; dashed red: fourth-order Fourier approximation. Dotted black: equilibrium leaflet positions. The boundary parameter is a measure of the distance around the boundary between the lipid-excluded surface of the protein and the membrane leaflet. (**C**) Solid and dotted blue: slopes of the membrane normal to the boundary. Dashed red: fourth-order Fourier slope approximations.

protein–membrane contact curve (*Figure 9A*) and on the far-field boundaries. Previously, we applied a similar membrane model in an attempt to predict the protein–membrane interface for M2 channels lacking the AH domains (*Zhou et al., 2020*); here, we augmented this approach by directly using the MD-generated membrane distortions at the protein boundaries to parametrized this boundary value problem.

The boundary conditions for the membrane–protein contact curve were taken directly from the MD simulations. We used a fourth-order Fourier series (red dash) to fit the displacement along the boundary (blue) as shown in *Figure 9B* for the parallel AH domain model (see *Appendix 1—figure 5* for 2L0J and 2N70 boundaries). The pronounced pinching in the upper leaflet is most evident in this plot as can be seen by comparing the difference between the MD displacement curve and the equilibrium height $+L_C/2$ (dashed curve). The slope along the inner boundary is much more noisy (solid and dotted blue curves) due to natural fluctuations as well difficulties in numerically taking derivatives of the mean membrane surfaces (*Figure 9C*). The fourth-order Fourier series approximation (red dashed curves) smooths the MD data reducing this noise and avoids imposing high-energy kinks on the boundary. Comparing panels B and C provides an idea of how membrane displacements are coupled to the membrane curvatures at the protein. Accurately solving the biharmonic equation requires knowledge of both of these features or knowledge of one and how it phenomenologically relates to the other. In our opinion, all-atom MD is currently the best approach for studying the coupling between both of these quantities. Most previous studies have either explicitly explored many different membrane slopes for a given displacement (*Nielsen et al., 1998*), or assumed (without justification) that the slope should be proportional to the membrane displacement from equilibrium (*Choe et al., 2008*), and to our knowledge, only Sodt and co-workers have analyzed all-atom MD to determine the coupling between membrane slope and local compression adjacent to the protein (*Sodt et al., 2017*).

Next, we numerically solved for the bilayer shape in a flat 200 Å × 200 Å patch, which is four times larger than the simulation patch size, using the inner boundary conditions taken from MD and the bilayer parameter values in *Table 2* for a 30% cholesterol bilayer. The far-field boundary conditions were clamped at zero slope and the upper and lower leaflets set to the equilibrium value $L_C/2 = \pm17.5$ Å, based on thickness values extracted from our simulations. The areal compression modulus ($K_a$) was obtained from measurements on similar membranes reported by *Henriksen et al., 2006*,

while the mean bending modulus ($K_C$) was determined from a consensus value for a pure POPC bilayer with a scaling argument again using $K_C$ values reported by Henriksen. We assumed the Gaussian bending modulus ($K_G$) scales linearly with $K_C$ following the simulation work of Deserno (**Hu et al., 2012**), and we use a moderate surface tension ($\alpha$) (**Latorraca et al., 2014**).

A side-by-side comparison of the upper and lower leaflet heights from the MD (left column) and the continuum calculations (right column) for all three configurations shows that the model preserves many features present in the MD surfaces (**Figure 10**). For instance, the upper leaflet shows significant curvature in the elastic calculations, while the lower leaflets are flatter with the twofold structures exhibiting a muted saddle-like shape present in the MD. The magnitude of the displacements is also similar in both cases, which is expected at the inner boundaries since they are identical along this curve. The spatial extent of the distortions between the MD and continuum model are both qualitatively and quantitatively similar, but the computed membrane heights decay to the equilibrium, far-field boundary value over a smaller distance than the MD. This is illustrated in the upper leaflet of 2L0J in panel A, where the edge of the 1 Å height contour takes a cross-like shape in the MD surface, likely because of stabilization from the periodic boundary conditions. For the continuum surface, the contour is instead more rounded and therefore less extended. Nonetheless, both methods produce smoothly varying heights, and their correspondence is remarkable.

Differences that do exist likely arise from several factors including the elastic model penalizing higher-order frequency modes, and insufficient averaging of the MD to gain adequate sampling and remove low-amplitude, high-frequency features. This latter point is critical when short MD simulations are performed, and it becomes impossible to interpret membrane shapes with limited all-atom data. However, the greatest difference between the MD and continuum solutions is likely that the MD has imposed periodic boundary conditions on a moderately sized membrane patch, while the continuum model has no imposed periodicity on a much greater region with flat, far-field boundaries twice as large in both linear dimensions. The deformation crosses the boundaries in each MD surface (left column), but not in the continuum results. Periodicity certainly impacts the surface shapes for the MD, and the continuum results provide a more realistic view of what an isolated protein experiences in the membrane.

We also computed the total membrane deformation energy in a flat bilayer for each of these conformations. Given the significant bilayer compression and induced upper leaflet curvature near the protein boundary, we predict that all three conformations induce significant elastic deformation energy in the bilayer with the fourfold 2L0J the lowest (36 kT), then the twofold 2N70 structure (89 kT), and the largest is the twofold parallel AH domain model (115 kT). While these values are substantial, they fall in the range of reported literature values for other computed membrane distortion energies induced by proteins such as 100 kT for lipid scrambling nhTMEM16 (**Bethel and Grabe, 2016**) and 50 kT for mechanosensitive gating of MscL **Ursell et al., 2007**; the latter of which is consistent with the experimental total gating energy of 51 kT (**Chiang et al., 2004**). Additionally, the high cholesterol composition of these membranes increases the bilayer thickness (35 Å) and all of the moduli, and if we reduce these values to a pure POPC bilayers (~28.5 Å), our model predicts significantly lower energy values that drop by more than half to 15, 39, and 47 kT for 2L0J, 2N70, and the parallel AH domain model, respectively (see **Appendix 1—figure 6**). While the twofold models induce higher membrane distortion energies, the total energy of the configuration is unknown since we have not quantified the energy of the other interactions in the system (protein enthalpy and entropy, protein–lipid interactions, protein–solvent interactions, etc.), and these contributions can be significant. We also want to point out that the membrane energies, while dominated by mean curvature and compression, also contain Gaussian curvature energy, since the membrane–protein contact curves are non-trivial, and the total surface is not topologically equivalent to a plane.

## C2 symmetry broken channels prefer NGC

We used the continuum model to predict whether different M2 conformations would prefer different membrane geometries as assessed by the total membrane-bending energy of different situations. To do this, we applied different far-field boundary conditions to impose global shapes on the membrane ranging from inward-budding, concave spherical caps to outward-budding, convex caps, which include flat membranes as the geometry transitions from concave to convex, as well as saddle geometries of pure NGC. We assumed that the principal curvatures ($\kappa_1$ and $\kappa_2$) for each background shape were equal

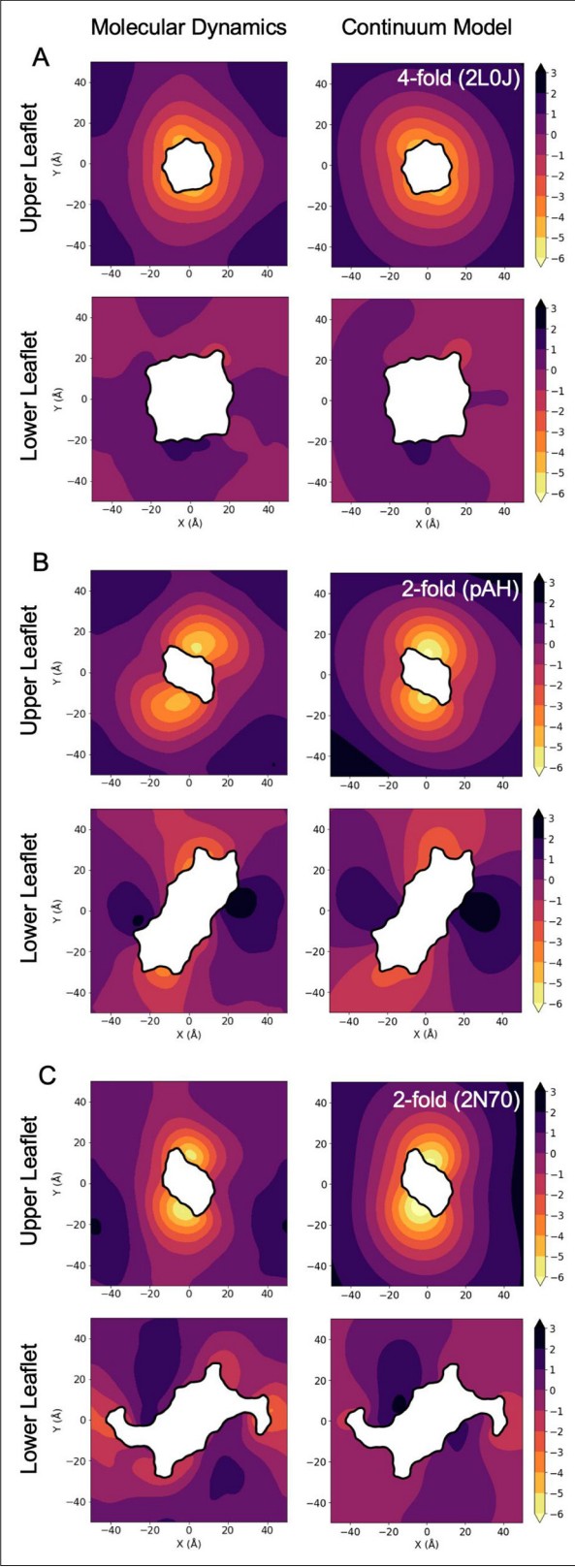

**Figure 10.** Continuum model membrane deformations compared to molecular dynamics (MD) surfaces. Left column: MD upper and lower leaflet mean positions. Right column: continuum model minimum energy upper and lower leaflet surfaces for a flat membrane (200 Å by 200 Å membrane with zero displacement and slope on the outer boundary – entire patch not shown), with inner boundary conditions at the protein taken from the MD surfaces in the left column. (**A**) 2L0J. (**B**) Parallel AH domain model (pAH). (**C**) 2N70.

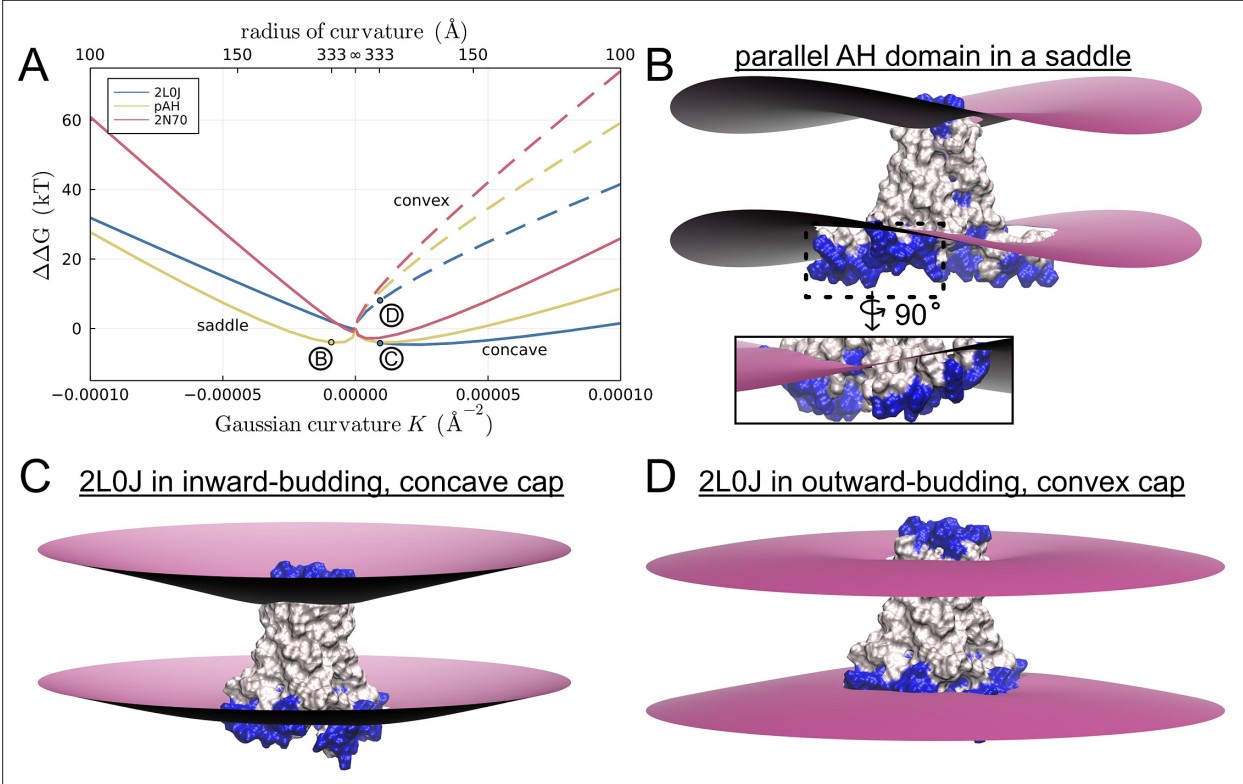

**Figure 11.** C2 symmetry broken conformations are stabilized in membranes with negative Gaussian curvature. (**A**) Transfer free energy $\Delta\Delta G(K)$ for moving M2 from a flat region to a region with Gaussian curvature $K = \pm 1/R_c^2$, where $R_c$ is the radius of curvature (top $X$-axis). Solid lines at negative $K$ show $\Delta\Delta G$ for three M2 models in saddles. Solid (dashed) lines at positive Gaussian curvature show $\Delta\Delta G$ in concave (convex) membranes. Labeled points B, C, and D on the plot indicate the energy for the membranes shown in panels B–D. For all shapes, $K_1 = \pm K_2$. (**B**) Parallel amphipathic helix (AH) domain model in a saddle. Inset: rotated close-up view of the lower leaflet distortion around the AH domain. (**C**) 2L0J in a concave spherical membrane. (**D**) 2L0J in a convex spherical membrane.

to each other (spherical caps) or opposite in sign (saddles), and we employed values corresponding to linear dimensions ranging from 100 Å radii of curvature ($R_c$) to infinite in the case of the flat plane. Next, we assumed that each conformation retained its flat-bilayer boundary conditions as it migrates into regions of different curvature, thus we used the MD-generated boundary conditions (**Figure 9**) on the inner boundary for all calculations. This assumption is based on the realization that failing to satisfy hydrophobic mismatch and lipid tilt at the membrane boundary can result in very large energy penalties, and the membrane adopts the same shape at the membrane–protein interface regardless of the global geometry. We note that the computed boundary conditions and membrane deflections were stable before and after the membrane repacking (see Appendix: Section 1), suggesting that the assumption is just. With these assumptions, we then computed the transfer free energy for each M2 configuration migrating from a region of flat curvature (i.e., the plasma membrane) into regions of different curvature, setting the reference membrane distortion in each case to zero for the flat reference case (**Figure 11**). The $X$-axis in panel A of this figure, therefore, represents a continuum of shapes with NGC saddles on the left-hand side, flat bilayers at zero, and concave or convex caps on the right-hand side. Transfer free energies were then computed by calculating the difference between the flat empty bilayer plus the total membrane elastic energy of an M2 configuration embedded in a curved space minus the energy of the empty curved surface plus the M2 configuration embedded in a flat membrane (see Appendix **Equation A2**). The flat empty bilayer in this cycle has no energy, and the protein does not contribute to any change in the Gaussian curvature energy since the inner boundary conditions are fixed. This calculation is akin to a constant-area Helmholtz free energy in which the lipids displaced by the protein as it enters a curved space flow into the flat region evacuated by the protein, hence relieving all their stored strain energy.

The C2-symmetric, parallel AH domain model is stabilized in NGC by 4 kT in a saddle with a 333 Å (33 nm) radius of curvature (gold curve at NGCs in *Figure 11A*). The figure and inset show how the protein aligns in the membrane with its AH domains pointing along the concave principal curvature direction. It remains stable up to curvatures close to 150 Å, which is the linear dimension of stalled viral buds. At this curvature, the fourfold 2L0J is moderately destabilized in NGC by about 2 kT, and the C2-symmetric 2N70 structure by 3 kT (blue and red curves at NGCs, respectively).

All three M2 configurations preferentially sort to inward-budding, spherical caps of concave curvature (solid lines at positive Gaussian curvatures). Stabilization in this geometry is related to the destabilization in the convex caps – the pinching and membrane slope induced by the protein on the upper leaflet matches the global curvature of the concave caps (see *Figure 11C* for 2L0J), hence reducing the total membrane deformation energy. This geometry is not relevant for a budding virus, but it does suggest that certain M2 channel configurations may become 'trapped' in cellular regions rich in concave curvature such as caveolae. The calculations reveal that all three proteins are excluded by the outward-budding, convex spherical cap of a budding virus. At a radius of 125 Å corresponding to stalled buds observed in EM, the energy values range from 30 to 52 kT higher than the energy in a flat bilayer (dashed lines in *Figure 11A*). These calculations corroborate the finding that M2 are not observed in the cap of stalled buds (*Rossman et al., 2010*). Panel D shows the solution for the fourfold 2L0J structure, and it is clear why the membrane-bending energy is high; the pinching and membrane slope imposed by the protein on the upper (extracellular) leaflet is opposite to the natural curvature of the convex spherical cap. Since upper leaflet pinching is a feature of all M2 configurations studied here, they all are penalized in this geometry.

As noted earlier, membrane–protein interactions depend on the biophysical properties of the membrane, and membrane composition in the cell is extremely heterogeneous. To begin to address how the bilayer properties influence curvature sensing, we recomputed the energy values in *Figure 11A* in a thinner (28.5 Å) and thicker (37 Å) membrane with additional properties representative of cholesterol-free and cholesterol enriched (50%) membranes, respectively (*Appendix 1—figure 6*). Parameters for these membranes are given in *Table 2*. The general shapes of the curves for saddle and outward-budding convex geometries are similar to those in *Figure 11A*; however, all of the energies are smaller in magnitude in the thinner, cholesterol-free membrane since the moduli are reduced and the membrane thickness is closer to the hydrophobic thickness of the protein reducing the relative amount of pinching in the upper leaflet. Additionally, the stabilization of the parallel AH model in NGC is diminished. The most notable difference is that proteins are now destabilized in thin, concave curved membranes compared to thick, cholesterol-rich membranes (*Appendix 1—figure 6*).

## Discussion

Our simulations of the fourfold symmetric 2L0J M2 channel reveal that the AH domains are highly dynamic and rearrange in the membrane, breaking both four- and twofold symmetry of the *entire* channel, while the TM portion experiences only smaller deviations. This mobility of the amphipathic helices appears robust across a range of leaflet tension and packing conditions, as it is seen in the initial unrestrained simulation (*Figure 2*) and in the repacked conditions (see Appendix: Section 1). Our restrained simulations highlight the need for proper balancing of leaflet packing in simulations with highly asymmetric proteins. Though the AH mobility and upper leaflet deformation are maintained, other aspects of the membrane behavior change substantially in response to packing differences. Upon matching the leaflet areas properly (*Figure 2*), we resolve much of the differential tension in the bilayer and generate leaflets that behave nearly identical to each other away from the protein.

Not only does the protein relax its conformation in the bilayer, but the membrane also adapts its shape around the channel. The repacked, restrained simulations in three different M2 configurations exhibit common features in the induced membrane distortion patterns. We observed pronounced pinching and large curvature in the extracellular leaflets, with a rotationally uniform pattern around 2L0J and twofold symmetric patterns around the parallel AH model and 2N70. The lower leaflet lipids in each simulation experienced greater tilt around the amphipathic helices, with lipid tails 'wrapping' around the partially inserted AH domains. Consequently, the increase in tilt decreases the vertical reach of the lower leaflet lipids, locally lowering the bilayer midplane, and the upper leaflet pinches and bends to accommodate. Thus, our work suggests that the AH domains cause upper leaflet bending and compression in patterns that reflect how the individual AH domains are oriented with

respect to the stable TM core. Both twofold symmetric conformations induce a twofold symmetric deformation that has more upper leaflet pinching and bending along one direction than another, while the fourfold symmetric 2L0J simulation creates a more radially symmetric pattern. This twofold symmetry is likely important for stabilization in saddle-like geometries, but it is not sufficient as 2N70 is not stable in NGC surfaces. The side-by-side packing of fully folded and inserted AH domains in the parallel AH model appears to enhance the membrane distortion compared to the partially unfolded AH in 2N70, producing stronger lipid tilts and an enhanced saddle-like deflection pattern that is minimized in background saddle shapes.

Our continuum calculations provide an assessment of the membrane distortion energy induced by each conformation in a flat bilayer, and we computed substantial values on the order of 36–115 kT in the thick, cholesterol-rich membranes (~35 Å) simulated here, but more moderate values (15–39 kT) in thinner, cholesterol-free membranes (~28.5 Å). In future work, we hope to verify these continuum predictions for the membrane energy using all-atom methodologies developed by the Faraldo-Gómez lab (*Fiorin et al., 2020*), and we are encouraged by studies that have shown that MD-derived energies agree semi-quantitatively with experiment on similar systems (*Sun et al., 2019*). Moreover, it would be informative to know the equilibrium probability of each of these three M2 conformations, but this would require computing the total free energy of the entire system including the protein, which is beyond our current scope. Nonetheless, these systems allowed us to address an important question – do different channel conformations prefer different membrane geometries? The answer is yes, as our elasticity calculations in different background curvatures revealed that the membrane energy has a complex dependence on the magnitude of the curvature and surface shape. Qualitatively our results can be understood in terms of the MD-generated distortion profiles in flat bilayers. The upper leaflet curvatures produced by all configurations favor migration into concave spherical regions and disfavor convex geometries, and the underlying mechanism is a relief of strain or increase in strain depending on the background geometry in a similar spirit to the curvature sensing model proposed by Johnson and co-workers (*Fu et al., 2021*). Thus, our work corroborates the finding that M2 is not enriched in the convex spherical cap of budding virions. The fourfold 2L0J structure does not favor saddle regions of moderate NGC, and despite both C2-symmetric conformations producing curvature distortions that appear compatible with saddles, only one is stabilized in NGC. Within the plane of the membrane the twofold symmetric proteins have a long and a short axis, and they align with the principal curvatures of the saddle to minimize the membrane elastic energy as shown in *Figure 11*. However, only the parallel AH domain model is stabilized in NGC saddles relative to a flat bilayer, while the 2N70 structure is not. This finding highlights the importance of the AH domains, their orientations, and the depth in the membrane. While the stabilization energy for the parallel AH model is modest, it could lead to enrichments in saddles over the flat regions of the plasma membrane by 50- to 400-fold depending on the exact lipid composition of the budding virion (see *Figure 11* and *Appendix 1—figure 6B*), and their presence would likely stabilize saddle formation.

There are a variety of M2 structures. We call attention to two distinguishing properties of these structures: the helicities of the amphipathic domains and the degrees of symmetry. 2L0J may represent M2 in a membrane environment without curvature, whereas our work follows others (*Elkins et al., 2017*) in supposing 2N70 may represent M2 in a highly curved environment, one possibly more curved than the physiological case. Indeed, many differences in M2 structures could be partially accounted for by considering the curvature of the membrane mimetic. C2 symmetry of the membrane deformation pattern appears energetically significant to M2's negative curvature sensitivity, but it may contribute to structural changes necessary to access the C2 conformations competent for budding. Another important consideration is that the viral membrane lipid composition is quite different from both the plasma membrane and the experimental composition we mimicked here. We simulated a bilayer containing PC, cholesterol, and negative PG, but the viral envelope has been shown to contain sphingolipids, have greater cholesterol content than what we considered, and be enriched in phosphatidylethanolamine (PE) and phosphatidylserine (PS) over PC and PG, respectively (*Gerl et al., 2012*; *Ivanova et al., 2015*). These changes create a robust virion that can better withstand harsh environments, while also retaining the fusogenic properties required for infection (*Gerl et al., 2012*). The increased cholesterol content would rigidify the membrane and enhance many of the energies reported here, but it is difficult to say how the sphingolipids, PS, and decreased headgroup size of PE

compared to PC would impact our results. Additional MD simulations with these compositions would be required to answer this question.

Compared to the CD constructs considered here, full-length M2 has an extended C-terminal domain that is disordered. While the structural implications are far from certain, it is likely that the full-length protein is even more dynamic in the entire C-terminal region to accommodate conformational entropy for the disordered tail. DEER studies (*Herneisen et al., 2017*) also suggest that full-length M2 may have a greater propensity to explore C2 conformations than the CD constructs. The dynamics observed here were already sufficient to complicate analysis of membrane deformation patterns around M2, so proper interpretation of simulation studies of full-length M2 presents a challenge for future work.

Understanding the role of M2 in viral egress could lead to novel therapeutic strategies not just for influenza, but other viruses or pathogens that bend mammalian plasma membranes. Most M2 drugs have historically been pore blockers, but drugs antagonizing C2 tetramerization at an external site may merit additional exploration. To our knowledge, no drugs are presently known to specifically inhibit M2 localization to NGC. Early M2 structures showed rimantadine binding at an external site, but it was later determined to be a lower-affinity site compared to the pore. Finding higher-affinity ligands for this alternate rimantadine site may yield additional insight into the drugability of M2's membrane-bending function. Additionally, cholesterol has been shown to bind at two of the tetramer interfaces (*Elkins et al., 2017*), suggesting that specific binding to this shallow groove is possible. However, it is not clear whether existing structural information is sufficient to find binders at the peripheral site: 2N70 likely represents an extreme C2 conformation, not the C2 conformation present on average at the budding neck. Further studies combining unrestrained MD with continuum energy calculations may be able to find dynamically accessible conformations with low energy in saddle-shaped membranes, and such conformations could be fruitful poses for docking or drug design, thus opening up the possibility for using small molecules to stabilize M2 channels in fission-incompetent conformations.

## Methods
### Molecular dynamics
#### Setup
Simulations of the M2 CD were initiated from PDBID 2L0J (residues 22–62) (*Sharma et al., 2010*), the parallel AH domain model based on PDBID 2N70 (residues 22–60), and PDBID 2N70 (residues 18–60) (*Andreas et al., 2015*). Residues were assigned standard protonation states at pH 7, with the exception of His37 which retained the protonation states from PDBID 2L0J (doubly protonated on chains A and C, epsilon protonated on chains B and D). Structures were embedded in 56:14:30 POPC:POPG:cholesterol bilayers, and solvated with 150 mM KCl resulting in a neutralized system using CHARMM-GUI (*Lee et al., 2016*). The force fields used for protein, lipids, and water were CHARMM36m (*Huang et al., 2017*), CHARMM36 (*Klauda et al., 2010*), and TIP3P (*Jorgensen et al., 1983*), respectively. Standard CHARMM parameters were used for ions (*Vanommeslaeghe et al., 2010*).

#### Production
Simulations were run on the Wynton HPC cluster with GPU nodes using GROMACS 2018 (*Abraham et al., 2015*). All simulations were minimized and equilibrated using the default options provided by CHARMM-GUI, excepting fully restrained simulations which retained backbone restraints with a force constant of 4000 kJ/mole/nm$^2$ throughout. Briefly, minimization was performed for 5000 steps with protein backbone, protein side chain, and lipid harmonic positional restraints at force constants of 4000, 2000, and 1000 kJ/mole/nm$^2$, respectively, as well as dihedral restraints with a force constant of 1000 kJ/mole/nm$^2$. A multi-step equilibration protocol stepped down the restraints over a series of 2 ns phases based on recommended CHARMM-GUI defaults. Following equilibration, an unrestrained series of pre-production simulations totaling 10 ns was run using a 2-fs timestep, either a Parrinello–Rahman barostat (*Parrinello and Rahman, 1981*) for all repacked simulations (#5–14 in *Table 1*) or Berendsen barostat (*Berendsen et al., 1984*) for the initial set (#1–4 in *Table 1*) with semi-isotropic pressure control at 1 atm, and a *Hoover, 1985*; *Nosé, 1984* set to 303.15 K. Nonbonded interactions

were cut off at 12 Å with force-switching between 10 and 12 Å, long-range electrostatics were calculated with particle mesh Ewald (*Darden et al., 1993*), and hydrogens were constrained with the LINCS algorithm (*Hess et al., 1997*). Final production MD proceeded using the same options with hydrogen mass repartitioning (*Balusek et al., 2019*) enabled to allow for use of a 4-fs timestep.

## Analysis

Systems were first centered on the protein and wrapped to make molecules whole, then rotated and translated (constrained to the *XY* plane) to maintain the starting configuration using GROMACS (gmx trjconv). Membrane surface calculations were performed using a custom analysis package based on MDAnalysis (*Michaud-Agrawal et al., 2011*), NumPy (*Harris et al., 2020*), and SciPy (*Virtanen et al., 2020*) with the same approach as outlined previously (*Bethel and Grabe, 2016*). Briefly, we erect a rectilinear grid with 1 Å spacing everywhere except at the protein–membrane interface, where we use a level set method based on the protein structure to move adjacent membrane grid points onto the surface (*Argudo et al., 2017*). Then, the positions of C22 and C32 atoms on POPC and POPG residues (first carbon atoms of each lipid tail) from MD simulations were interpolated onto this distorted grid using SciPy's implementation of the Clough–Tocher scheme to construct, for every timepoint analyzed, a hydrophobic surface for each leaflet. To avoid edge effects, the simulation frames were expanded by mirroring lipids within 18 Å of the box edge, producing an expanded box of 126 Å × 126 Å over which grids were interpolated and then trimmed back down to the true box size. This produces interpolated grids that vary smoothly over the entire 95 Å × 95 Å grid and capture the full periodicity of the system. Bilayer hydrophobic thicknesses were then calculated by taking the difference between interpolated upper and lower leaflet hydrophobic surfaces. All surfaces were then averaged across timepoints. These profiles were visualized using MATLAB R2015R [MathWorks, Natick, MA, USA], Matplotlib (*Hunter, 2007*), Seaborn (*Waskom, 2021*), Plots.jl (*Christ et al., 2023*), and Julia (*Bezanson et al., 2012*). Membrane midplanes were calculated and plotted following the same procedure with the terminal carbon atoms to generate tail surfaces for each leaflet, which were then averaged together to generate midplane surfaces for each timestep. Bilayer hydrophobic thicknesses and leaflet thicknesses were then calculated as the absolute difference between the relevant surfaces.

Lipid tilt angles were calculated per simulation frame for every phospholipid. The tilt angle was defined as the angle between the *Z*-axis of the box, taken as the bilayer normal, and the vector pointing from the phosphorous atom of each lipid phosphate group to the midpoint between the last carbon atoms of each lipid tail. We interpolated a tilt surface using the same grid and method as for the membrane surfaces just described. Tilt surfaces for each timepoint are then averaged together across the full production time. Average membrane properties were computed over the second half of each trajectory. See *Table 1* for a list of simulation times.

Lateral pressure profiles and leaflet tensions were calculated as follows. A 100-ns extension simulation was performed from the end of each production run, and positions and velocities were saved every 10 ps. Next, the Gromacs-LS package was used to calculate the three-dimensional stress tensor from this data on a 1.0-Å cubic grid. We computed the lateral pressure in the *XY* plane of the membrane from the stress tensor components as follows: $p_L = - (\sigma_{xx} + \sigma_{yy})/2$, ignoring anisotropies that may result from the protein. Additionally, we assumed that the pressure normal to the membrane is the negative of the *Z*-dimensional component of the stress tensor: $p_N = -\sigma_{zz}$. Each of these is further assumed to only depend on the *z* coordinate. The lateral pressure profile is defined as the difference of the lateral and normal pressures, $\pi(z) = p_L(z) - p_N(z)$, and is integrated from the midplane center to the top and bottom of the simulation box to obtain the tension in the upper and lower leaflets, respectively. To define the membrane midplane, we used the Gromacs density function to calculate the mass fraction of the upper and lower leaflet lipids over the same three-dimensional grid as the stress tensor was calculated. For every (*X*,*Y*) coordinate over the grid, the upper and lower leaflet tensions were calculated based on the local *z* coordinate of the membrane midplane. These upper and lower leaflet tensions were then averaged over the (*X*,*Y*) plane, excluding the region occupied by the protein, to get the reported values.

## Continuum membrane mechanics model

Continuum elastic numerical calculations were performed on 200 × 200 Å bilayer patches, which are sufficiently large to allow local distortions at the protein to relax (decay length $\lambda_D = (1/\gamma)^{\frac{1}{2}} \sim 60$ Å,

where $\gamma = \alpha/K_C$), while being small enough to capture curvatures relevant to viral budding. Smaller patches induced higher energetic penalties, while increasing the patch size did not significantly reduce the elastic energy. The protein–membrane boundary positions were taken from the intersections of the mean MD membrane surfaces with the lipid-excluded surface of the proteins (i.e., the molecular surfaces using a 4.58-Å probe radius). The inner slope boundary conditions were set to the normal component of the gradient of the MD mean surfaces. Outer boundary conditions were set to impose a flat bilayer, a spherical cap, or saddle background. With and without protein inclusions, for spherical geometry, the outer boundary upper $u^+$ and lower $u^-$ leaflet displacements were set to:

$$u^+ = u^- = \frac{2}{\gamma R_C}\left(\, I_0\left(\sqrt{\gamma}r\right) - 1\right)$$

while for saddles, the displacements were set to:

$$u^+ = u^- = \frac{4}{\gamma R_C}I_2(\sqrt{\gamma}r)\cos(2(\theta + \delta))$$

where $I_0$ and $I_2$ are modified Bessel functions of the first kind, $r$ and $\theta$ are polar coordinates, and $\delta$ is an arbitrary constant (see **Appendix 1—figure 7** and Appendix: Section 3). The outer boundary slopes were set to the boundary-normal component of the gradients of the above expression. In an empty membrane, with the above boundary conditions, these Bessel function solutions are solutions of the Euler–Lagrange equations throughout the membrane, and our numerical solver produced surfaces of these forms. The small $r$ limiting forms indicate that near the origin, these boundary conditions produce surfaces with Gaussian curvatures of $K = \pm 1/R_c^2$.

We calculated the difference in minimum energy configurations between membranes with and without M2 for surfaces with a range of Gaussian curvatures. For simplicity, we only considered surfaces whose principal curvatures $\kappa_1$ and $\kappa_2$ were the same magnitude: $|\kappa_1| = |\kappa_2| = 1/R_c$. The only free parameters were a constant $z_0$ offset and the orientation of the saddle $\delta$. For spherical caps, $z_0$ was optimized using Brent's method. For saddles, $z_0$ and $\delta$ were optimized using the Nelder–Mead method. Searches were implemented in Julia using Optim (**Mogensen and Riseth, 2018**).

## Parameter estimates for the continuum model

Membrane parameter values were estimated based on cholesterol doped POPC bilayers, ignoring the small additional amounts of POPG used in our simulations. In **Table 2**, where a reference is cited the value was taken directly from the paper, and in other cases, we had to extrapolate from existing data as described next.

### Membrane thickness ($L_C$)

Ferreira and co-workers provide NMR-derived tail lengths for POPC with increasing amounts of cholesterol, getting 14.5 Å at 34% cholesterol and 15.3 Å at 50% cholesterol (**Ferreira et al., 2013**). Treating the 5.5% difference going from the lower to higher cholesterol condition and adding that to our MD value of 35 Å bilayer thickness at 30% cholesterol, our 50% cholesterol estimate is 37 Å.

### Mean bending modulus ($K_C$)

The $K_C$ value for a pure POPC membrane was obtained from averaging five reported values obtained using four different methods: all-atom MD (**Drabik et al., 2020**; **Venable et al., 2015**), small-angle X-ray (**Kučerka et al., 2006**), micropipette aspiration (**Henriksen et al., 2006**), and flicker noise spectroscopy analysis (**Drabik et al., 2020**). We estimated the mean bending modulus at 30% cholesterol using the scaling from 0% to 30% cholesterol from Henriksen, which is 225% (**Henriksen et al., 2006**), along with the mean value at 0% reported in the table. For cholesterol content above 30 %, Pan and co-workers showed that the bending modulus is rather constant for a single-chain mono-unsaturated lipid SOPC, where they reported a 4% increase from 30% to 50% (**Pan et al., 2009**), which we use here.

## Gaussian bending modulus ($K_G$)

We use the phenomenological scaling relation between $K_C$ and $K_G$ reported by Hu ($K_G = -0.9K_C$) to arrive at $K_G$ from the values reported for $K_C$ (*Hu et al., 2012*). This model is based on a coarse-grained Cooke model of the lipids, and we note that other all-atom approaches to determine this relationship arrive at smaller ratios (*Venable et al., 2015*). Nonetheless, while we include this parameter in our table for completeness, it actually has no impact on the free energy values reported here because the total Gaussian curvature is constant in these cases.

## Areal compression modulus ($K_a$)

We assume that changes in $K_a$ above 30% cholesterol plateau as observed for $L_C$ and $K_C$, and we estimate only a 5 % increase from 30% to 50% cholesterol as reported in *Table 2*.

## Acknowledgements

We would like to thank Charles Wolgemuth, Bill DeGrado, Yongcheng Zhou, Kathleen Howard, Matthew Jacobson, and members of the Grabe lab for helpful discussions regarding this work. This work was supported by funds from the UCSF Biophysics Program, NIH postdoctoral fellowship to JL (4T32HL007731-28), NSF grant P0558710, and NIH grants R01GM117593 and R01GM137109. Hardware for simulations was provided by NIH R01GM089740.

---

# Additional information

### Funding

| Funder | Grant reference number | Author |
|---|---|---|
| National Institutes of Health | 4T32HL007731-28 | James Lincoff |
| National Science Foundation | P0558710 | Michael Grabe |
| National Institutes of Health | R01GM117593 | Michael Grabe |
| National Institutes of Health | R01GM137109 | Michael Grabe |
| National Institutes of Health | R01GM089740 | Michael Grabe |

The funders had no role in study design, data collection, and interpretation, or the decision to submit the work for publication.

### Author contributions

James Lincoff, Formal analysis, Funding acquisition, Validation, Investigation, Visualization, Methodology, Writing - original draft, Writing – review and editing; Cole VM Helsell, Conceptualization, Data curation, Formal analysis, Investigation, Visualization, Methodology, Writing - original draft, Writing – review and editing; Frank V Marcoline, Software, Formal analysis, Validation, Investigation, Visualization, Methodology, Writing - original draft, Writing – review and editing; Andrew M Natale, Formal analysis, Validation, Visualization, Writing – review and editing; Michael Grabe, Conceptualization, Resources, Software, Formal analysis, Supervision, Funding acquisition, Validation, Investigation, Visualization, Methodology, Writing - original draft, Project administration, Writing – review and editing

### Author ORCIDs

James Lincoff ⓘ https://orcid.org/0000-0003-3090-1240
Frank V Marcoline ⓘ http://orcid.org/0000-0002-8842-5470
Michael Grabe ⓘ https://orcid.org/0000-0003-3509-5997

### Decision letter and Author response

Decision letter https://doi.org/10.7554/eLife.81571.sa1

Author response https://doi.org/10.7554/eLife.81571.sa2

## Additional files

### Supplementary files
• MDAR checklist

### Data availability

The original DEER data reported in Figure 7 of *Kim et al., 2015* appears in panel B of *Figure 2* (green curves). The raw data is included as *Figure 2—source data 1*. This study used two previously published structures of M2 (PDBIDs 2L0J and 2N70) as well as a custom-built parallel AH domain construct based off of 2N70. The parallel AH domain construct structure is included as *Figure 4— source data 1*. Both source data files can be freely downloaded from the following link https://doi. org/10.5281/zenodo.6846507. Additionally, full-size images of the equilibrated all-atom snapshots from the protein-restrained simulations in *Figure 7* are included as source data *Figure 7—source data 1* – full size image of lipids around 2L0J, *Figure 7—source data 2* – full size image of lipids around parallel AH domain model, and *Figure 7—source data 3* – full size images of lipids around 2N70.

The following dataset was generated:

| Author(s) | Year | Dataset title | Dataset URL | Database and Identifier |
|---|---|---|---|---|
| Helsell CVM, Marcolie FV, Lincoff J, Natale AM, Grabe M | 2022 | Source data for "Membrane curvature sensing and symmetry breaking of the M2 proton channel from Influenza A" | https://doi.org/10.5281/zenodo.6846507 | Zenodo, 10.5281/zenodo.6846507 |

The following previously published datasets were used:

| Author(s) | Year | Dataset title | Dataset URL | Database and Identifier |
|---|---|---|---|---|
| Sharma M, Yi M, Dong H, Qin H, Peterson E, Busath DD, Zhou HX, Cross TA | 2010 | Solid State NMR structure of the M2 proton channel from Influenza A Virus in hydrated lipid bilayer | https://www.rcsb.org/structure/2L0J | RCSB Protein Data Bank, 2L0J |
| Andreas LB, Reese M, Eddy MT, Gelev V, Ni Q, Miller EA, Emsley L, Pintacuda G, Chou JJ, Griffin RG | 2015 | Two-fold symmetric structure of the 18-60 construct of S31N M2 from Influenza A in lipid bilayers | https://www.rcsb.org/structure/2N70 | RCSB Protein Data Bank, 2N70 |

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

## Appendix 1

### Section 1: Bilayer repacking simulations

The initial set of restrained simulations conducted for this study were generated directly from CHARMM-GUI with 200 lipids in the upper leaflet and 150 in the lower leaflet, each with 56:14:30 POPC:POPG:CHOL molar composition. We noted strong deflections in the upper leaflet surface, similar to the final repacked results presented in the main text (*Figures 3–5*), along with a nearly flat lower leaflet surface (*Figure 3C*). We also observed relatively low levels of lipid tilt in the upper leaflet, with elevated levels of tilt throughout the lower leaflet but enhanced near the protein. Initially, we attributed these leaflet asymmetries purely to the presence of the protein and its influence on the local membrane shape. It was noted during peer review, however, that the persistence of elevated tilt in the lower leaflet relative to the upper leaflet, even far from the protein at the box edge, might reflect an imbalance in the total leaflet packing with an underpacked, overthinned lower leaflet and an overpacked, compressed upper leaflet.

To address this concern, we calculated the lateral pressure profiles with the Gromacs-LS methodology (see Methods) for the original restrained 2L0J simulation (*Table 1*, simulation #2) revealing tension values of −10.8 mN/m in the upper leaflet and 24.6 mN/m in the lower leaflet. These values indicate that the upper leaflet is compressed while the lower leaflet is in tension. A priori, we do not know if these values are inconsistent with a deformed membrane exhibiting bending moments; however, the values are larger than most reported in the literature. Thus, we were concerned that the observed membrane deformations may be an artifact of poorly packed initial bilayers. In a natural bilayer, this area mismatch could be resolved through lipid exchange between leaflets or lateral lipid flow from a large pool of reserve lipids. In simulation, however, the small physical dimensions of the periodic box coupled with short simulation times prevent the relaxation of differential stress build up through either of these two mechanisms.

To correct for this mismatch, we start by assessing the expected leaflet areas of the simulation. All simulations were constructed using the CHARMM-GUI membrane builder, which can take aligned transmembrane (TM) protein structures from the Orientation of Proteins in Membranes (OPM) database and build leaflets of a desired lipid composition around them using an area-matching approach. For 2L0J, the upper leaflet protein area is 540 $\text{Å}^2$ and the lower leaflet protein area is 1560 $\text{Å}^2$. The initial generated setup, with 200 lipids in the upper leaflet and 150 in the lower leaflet, had expected total leaflet areas of 12,300 $\text{Å}^2$ for the upper leaflet and 10,400 $\text{Å}^2$ for the lower leaflet, producing a strong mismatch in the total upper and lower areas (protein plus lipid). We therefore built a series a simulations that reduced 25%, 50%, 75%, and 100% of the initial area mismatch by adding an additional 8, 16, 24, and 33 lipids to the lower leaflet, respectively. The upper leaflet was not changed.

These new simulations, termed Repack 1, 2, 3, and 4, respectively (*Table 1*, simulations #5–8), were run using the same initial restrained 2L0J simulation protocol and assessed for their leaflet properties and tensions. Tensions were calculated as before from 100 ns extension simulations following 2 µs of production using the Gromacs-LS package. Selected leaflet properties for each simulation are summarized in *Appendix 1—table 1*. Leaflet surfaces and bilayer thicknesses are plotted in *Appendix 1—figure 1*, leaflet tilt surfaces are plotted in *Appendix 1—figure 2*, and leaflet thicknesses are plotted in *Appendix 1—figure 3*.

The area per lipid (APL) trends (*Appendix 1—table 1*) clearly show the influence of changing the leaflet packing. In the original simulation, the overpacked upper leaflet APL is significantly smaller than the lower leaflet APL, likely from the large compression in the upper leaflet. As more lipids are added to the lower leaflet, the APL values of the two leaflets converge until they are within 1 $\text{Å}^2$ for Repack simulations 3 and 4. The same convergence is observed for the calculated leaflet thicknesses at the box edge – in the original simulation the upper leaflet is 2 Å thicker than the lower leaflet, but this difference is only 0.2 Å in Repack 4. This effect is also visible in the plots of the leaflet thicknesses (*Appendix 1—figure 3*), where the color ranges far from the protein become more similar with increased lipid packing. Likewise, the lipid tilt surfaces (*Appendix 1—figure 2*) show convergence of both bilayers far-away from the protein when the packing is adjusted. In contrast, the surface deflections (*Appendix 1—figure 1*) of both the upper and lower leaflets are fairly robust across repacking conditions, with strong curvature in the upper leaflet and a more flat lower leaflet.

The bilayer and leaflet tensions also decrease in magnitude with repacking, from an initial 13.8 mN/m bilayer tension (sum of upper and lower leaflet tensions) in the original simulation compared to 4.2 mN/m in Repack 4. This indicates that there is some stress even in the final repacked condition. We note, however, that the conical shape of 2L0J imposes curvature on the bilayer midplane (see *Figure 6*), which complicates the calculation of the lateral pressure profile as it relies on calculating the three-dimensional stress tensor. Some error in calculation is to be expected then, because the midplane curvature produces leaflets whose structures, and therefore calculated lateral and normal stresses, are not perfectly aligned along the *X*, *Y*, and *Z* axes. An adaptation of the Gromacs-LS implementation of this calculation, in order to calculate the true normal and lateral stresses accounting for the bilayer curvature, is beyond the scope of this work. Furthermore, while it is possible that some of the remaining calculated stress could be resolved with additional packing in the lower leaflet beyond the Repack 4 condition, we argue that the near-identical lipid behavior between the two leaflets as reflected in the APL and leaflet thicknesses is proof that we have sufficiently relaxed out the differential stress to the point that the membrane structure – which is the key in this manuscript, for defining the boundary conditions applied in the continuum membrane model – is not affected.

All additional production runs for 2L0J were initiated from the Repack 4 condition, with 200 lipids in the upper leaflet and 183 lipids in the lower leaflet (*Table 1*, simulation #8). We conducted repacked simulations of the parallel amphipathic helix (AH) domain model and 2N70 using the same corrected lipid packing (*Table 1*, simulations #13 and 14). We also ran a series of four unrestrained simulations of 2L0J (*Table 1*, simulations #9–12), initialized from the 2, 3, 4, and 5 µs timepoints of the 2L0J Repack 4 simulation to assess the impact of repacking on AH domain mobility. RMSD values for the AH and TM domains, as well as distance distributions for double electron–electron resonance (DEER) probe residues are plotted in *Appendix 1—figure 1*. The results are qualitatively similar to the initial unrestrained simulation (*Figure 2*), with high AH mobility compared to the TM domain and sampling of residue–residue distances that approach the maximum values of the DEER distributions. Thus, the large lower leaflet tension present in the original simulation was not the cause of the conformational heterogeneity in the AH domains. Though there are clear changes in the leaflet thicknesses and lipid tilts (*Appendix 1—figures 2 and 3*), it appears overall surfaces of the bilayer, and the mobility of the AH within them, are more stable across changes to the leaflet packing in the simulated systems.

## Section 2: Model of curvature sensing

The geometric shape and hydrophobic mismatch between a TM protein and the surrounding lipid bilayer can distort the shape of the membrane and change the total membrane energy. The protein-induced membrane deformation energy may be lower in some curved membrane regions than others, leading to curvature sensing. Using our numerical continuum elasticity solver (*Argudo et al., 2016*), we explored which background membrane curvatures are energetically favorable to M2, based on the deformations of lipid bilayers around M2 observed in molecular dynamics (MD) simulations.

Let $G_p(\kappa_1, \kappa_2)$ be the membrane elastic energy around a protein in a membrane region with principal curvatures $\kappa_1$ and $\kappa_2$, and compare this to the membrane elastic energy $G_e(\kappa_1, \kappa_2)$ of a similarly curved membrane region without protein inclusions:

$$\Delta G(\kappa_1, \kappa_2) = G_p(\kappa_1, \kappa_2) - G_e(\kappa_1, \kappa_2). \tag{A1}$$

To compare the protein-induced membrane energy difference in a curved region to that in a flat region ($\kappa_1 = \kappa_2 = 0$), we calculate $\Delta\Delta G(\kappa_1, \kappa_2) = \Delta G(\kappa_1, \kappa_2) - \Delta G(0, 0)$:

$$\begin{aligned} \Delta\Delta G(\kappa_1, \kappa_2) \quad &:= \quad \Delta G(\kappa_1, \kappa_2) - \Delta G(0, 0) \\ &= (G_p(\kappa_1, \kappa_2) - G_e(\kappa_1, \kappa_2)) - (G_p(0, 0) - G_e(0, 0)) \\ &= (G_p(\kappa_1, \kappa_2) + G_e(0, 0)) - (G_p(0, 0) + G_e(\kappa_1, \kappa_2)). \end{aligned} \tag{A2}$$

That is, $\Delta\Delta G(\kappa_1, \kappa_2)$ is the change in membrane energy when a protein is moved from flat region to a curved region, while simultaneously moving the displacing lipids from the curved region into the resulting hole in the flat region. Elsewhere we will refer to $\Delta\Delta G(\kappa_1, \kappa_2)$ as $\Delta\Delta G(K)$, where $K$ is the Gaussian curvature: $K = \kappa_1\kappa_2$.

## Section 3: Minimum energy surfaces

To set up numerical calculations in regions of prescribed curvature, we found analytic minimum energy solutions for protein-free membranes of chosen curvature, and used them as boundary conditions for our numerical solver. In a protein-free region, the Euler–Lagrange equations for the vertical displacements from equilibrium of the upper and lower membrane leaflets, $u^+$ and $u^-$, respectively, of an intrinsically flat membrane bilayer are

$$\nabla^4 u^+ - \gamma\nabla^2 u^+ + \beta(u^+ - u^-) = 0, \text{ and} \tag{A3}$$

$$\nabla^4 u^- - \gamma\nabla^2 u^- + \beta(u^- - u^+) = 0, \tag{A4}$$

where $\gamma = \alpha/K_c$ is the ratio of the stretch modulus $\alpha$ to bending modulus $K_c$, and $\beta = 2K_a/K_c/L_c^2$ is the ratio of the compression modulus $K_a$ scaled by the membrane thickness $L_c$ squared to the bending modulus. Solving these fourth-order differential equations requires specifying two sets of boundary conditions for each leaflet. We chose to specify $u^+$ and $u^-$, and their normal derivatives $\nabla u^+ \cdot \hat{n}$ and $\nabla u^- \cdot \hat{n}$, where $\hat{n}$ is the direction normal to the boundary.

Solutions to these equations are well known. For example, **Nielsen et al., 1998** solved the case of equal magnitude but opposite sign displacements and slopes. We instead looked at solutions with identical offsets and slopes on the boundaries. When $u^+ = u^- = u$, and $\nabla u^+ \cdot \hat{n} = \nabla u^- \cdot \hat{n} = \nabla u \cdot \hat{n}$ everywhere on the boundary, the two leaflets decouple, and minimal surfaces are solutions of the homogeneous equation

$$\nabla^4 u - \gamma\nabla^2 u = 0. \tag{A5}$$

In polar coordinates, the solution to this simpler biharmonic equation is

$$u(r, \phi) = C + D\ln(r) + \sum_{m=0}^{\infty}\left[A_m I_m(\sqrt{\gamma}r)\cos(m(\phi + \delta_m^I)) + B_m K_m(\sqrt{\gamma}r)\cos(m(\phi + \delta_m^K))\right], \tag{A6}$$

where $I_m$ and $K_m$ are order $m$ modified Bessel functions of the first and second kind, and $C$, $D$, $A_m$, $B_m$, $\delta_m^I$, and $\delta_m^K$ are constants determined by the boundary conditions. We were interested in exploring solutions which are finite at the origin and look either spherical ($m = 0$) or saddle like ($m = 2$).

### Spherical caps

A hemispherical cap touching the origin, centered at $z = R$ above the origin, has height $z_{cap}$:

$$\begin{aligned} z_{cap}(r) &= R - \sqrt{R^2 - r^2} \\ &= \frac{r^2}{2R}\left(1 + \frac{r^2}{2R^2} + \mathcal{O}(\frac{r^4}{R^4})\right) \\ &\approx \frac{r^2}{2R} \text{ for } r \ll R. \end{aligned} \tag{A7}$$

The Maclaurin series for $I_0(\sqrt{\gamma}r)$ is

$$I_0(\sqrt{\gamma}r) = 1 + \frac{\gamma r^2}{4} + \frac{\gamma^2 r^4}{64} + \mathcal{O}((\sqrt{\gamma}r)^6), \tag{A8}$$

so an empty membrane patch which looks locally like a spherical cap with radius of curvature $R$ is given by the $m = 0$, rotationally symmetric membrane with solution

$$\begin{aligned} u_{cap}(r, \phi) &= \frac{2}{\gamma R}\left(I_0(\sqrt{\gamma}r) - 1\right) \\ &\approx \frac{r^2}{2R} \text{for } \gamma r^2 \lesssim 1. \end{aligned} \tag{A9}$$

With $\alpha = 3 \times 10^{-13}$ N/Å and $K_c = 1.1 \times 10^{-9}$ N Å, $\gamma = \alpha/K_c = 2.7 \times 10^{-4}$ Å$^{-2}$. The membrane is expected to deviate from a spherical cap when $r \sim \sqrt{1/\gamma} = 60.2$ Å. **Appendix 1—figure 7A** shows the ideal mathematical spherical cap and the biharmonic solution used to simulate a spherical cap.

## Saddles

One definition of a unit saddle is $z = (x^2 - y^2) = \frac{1}{2}r^2\cos(2\phi)$. This saddle has principal curvatures $\kappa_1 = -\kappa_2 = 1$. To match our solutions above, we add in an arbitrary rotation by angle $\delta$, and scale the principal curvatures by $\kappa_1 = -\kappa_2 = 1/R$:

$$z_{\text{saddle}} = \frac{r^2}{2R}\cos(2(\phi + \delta)), \tag{A10}$$

where as $R \to \infty$ the surface becomes flat. The $m = 2$ saddle-like membrane deformation from *Equation A6* is

$$
\begin{aligned}
u_{\text{saddle}}(r, \phi) &= A_2\, \text{I}_2(\sqrt{\gamma}r)\cos(2(\phi + \delta_2^{\text{I}})) \\
&= A_2\left[\frac{\gamma r^2}{8} + \frac{\gamma^2 r^4}{96} + \mathcal{O}((\sqrt{\gamma}r)^6)\right]\cos(2(\phi + \delta_2^{\text{I}}))
\end{aligned} \tag{A11}
$$

Setting $A_2 = \frac{4}{\gamma R}$ and $\delta_2^{\text{I}} = \delta$ gives a minimum energy saddle-like surface such that $u_{\text{saddle}} \approx z_{\text{saddle}}$ when $\gamma r^2 \lesssim 1$. *Appendix 1—figure 7B* shows a cut along $\phi = -\delta$ of the ideal mathematical saddle and the biharmonic solution used to simulate a saddle, where $z_{\text{saddle}}$ and $u_{\text{saddle}}$ are extremal.

## External boundary conditions

For the boundary conditions along the protein–membrane contact curve, we used the values derived from MD simulations (*Appendix 1—figure 5*), and assumed that these boundary conditions do not significantly change as the curvature of the surrounding membrane patch changes, since deviations of the local boundary conditions will result in high-energy penalties in the hydrophobic, electrostatic, and steric energies.

For spherical caps of radius $R$, the external boundary conditions were set to

$$u^+ = u^- = u_{\text{cap}} = \frac{2}{\gamma R}\left(\text{I}_0(\sqrt{\gamma}r) - 1\right), \text{ and} \tag{A12}$$

$$\nabla u^+ \cdot \hat{n} = \nabla u^- \cdot \hat{n} = \nabla u_{\text{cap}} \cdot \hat{n} = \frac{2}{\sqrt{\gamma}R}\text{I}_1(\sqrt{\gamma}r)\,\hat{r} \cdot \hat{n}, \tag{A13}$$

where $\hat{r}$ is the unit vector in the radial direction. The sign of the radius of curvature determines if the cap is convex or concave from the extracellular side. For saddles with principal curvatures $1/R$ and $-1/R$, the external boundary conditions were set to

$$u^+ = u^- = u_{\text{saddle}} = \frac{4}{\gamma R}\text{I}_2(\sqrt{\gamma}r)\cos(2(\phi + \delta)), \text{ and} \tag{A14}$$

$$
\begin{aligned}
\nabla u^+ \cdot \hat{n} = \nabla u^- \cdot \hat{n} &= \nabla u_{\text{saddle}} \cdot \hat{n} \\
&= \frac{2}{\sqrt{\gamma}R}\left[\text{I}_1(\sqrt{\gamma}r) + \text{I}_3(\sqrt{\gamma}r)\right]\cos(2(\phi + \delta))\,\hat{r} \cdot \hat{n} \\
&\quad - \frac{8}{\gamma r R}\text{I}_2(\sqrt{\gamma}r)\sin(2(\phi + \delta))\,\hat{\phi} \cdot \hat{n}.
\end{aligned} \tag{A15}
$$

## Section 4: Elastic energy in a thinner membrane

The mean hydrophobic core thickness far from the protein in the three MD simulations, based on the POPC and POPG C22 and C23 atoms, was 35 Å. Therefore we used 35 Å for the continuum membrane solver. In *Appendix 1—figure 6* we reproduce the energetic profiles in *Figure 11*, but for membranes with core thickness $L_c = 28.5$ Å and 35 Å. As in the main text, we set the membrane-bending modulus $K_c = L_c^2/24 \times K_a$. The membrane energy for the thinner membrane was then minimized again over the free parameters ($z_0$ and $\delta$).

**Appendix 1—table 1.** Leaflet properties for restrained 2L0J simulations with varying lipid numbers.

| Simulation | Area per lipid (Å²) | | Thickness at box edges (Å) | | Tension (mN/m) | |
|---|---|---|---|---|---|---|
| (lower lipid #) | Upper leaflet | Lower leaflet | Upper leaflet | Lower leaflet | Upper leaflet | Lower leaflet |
| Original (150)* | 42.5 | 49.9 | 18.5 | 16.3 | −10.8 | 24.6 |
| Repack 1 (158) | 43.8 | 49.0 | 18.2 | 15.8 | −13.1 | 20.2 |
| Repack 2 (166) | 44.4 | 47.4 | 17.9 | 16.4 | −11.8 | 13.2 |
| Repack 3 (174) | 45.2 | 46.1 | 17.8 | 17.0 | −9.6 | 7.6 |
| Repack 4 (183)† | 46.4 | 45.1 | 17.4 | 17.2 | −3.8 | 8.0 |

*The original restrained 2L0J simulation used a semi-isotropic Berendsen barostat for pressure coupling. All repacked simulations, and extension simulations for Gromacs-LS calculations, use a Parrinello–Rahman barostat. Other simulation parameters were identical for all simulations.

†Lipid ratio used for all subsequent 2L0J analysis.

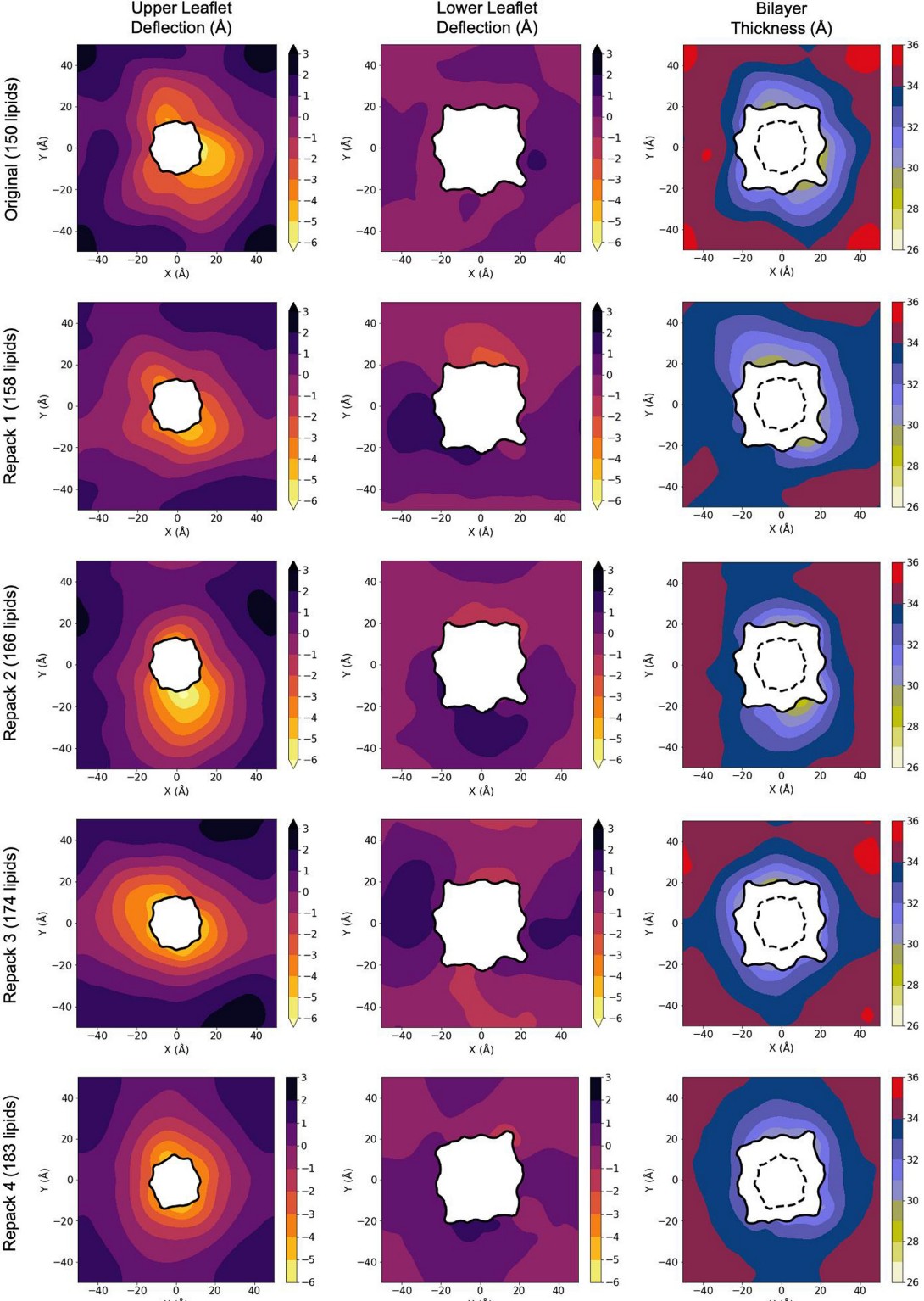

**Appendix 1—figure 1.** Leaflet surfaces (left and middle columns) and bilayer thicknesses (right column) for the original restrained 2L0J simulation and restrained Repack simulations 1–4. The upper leaflet has 200 lipids, and the lower leaflet lipid count is indicated in parentheses.

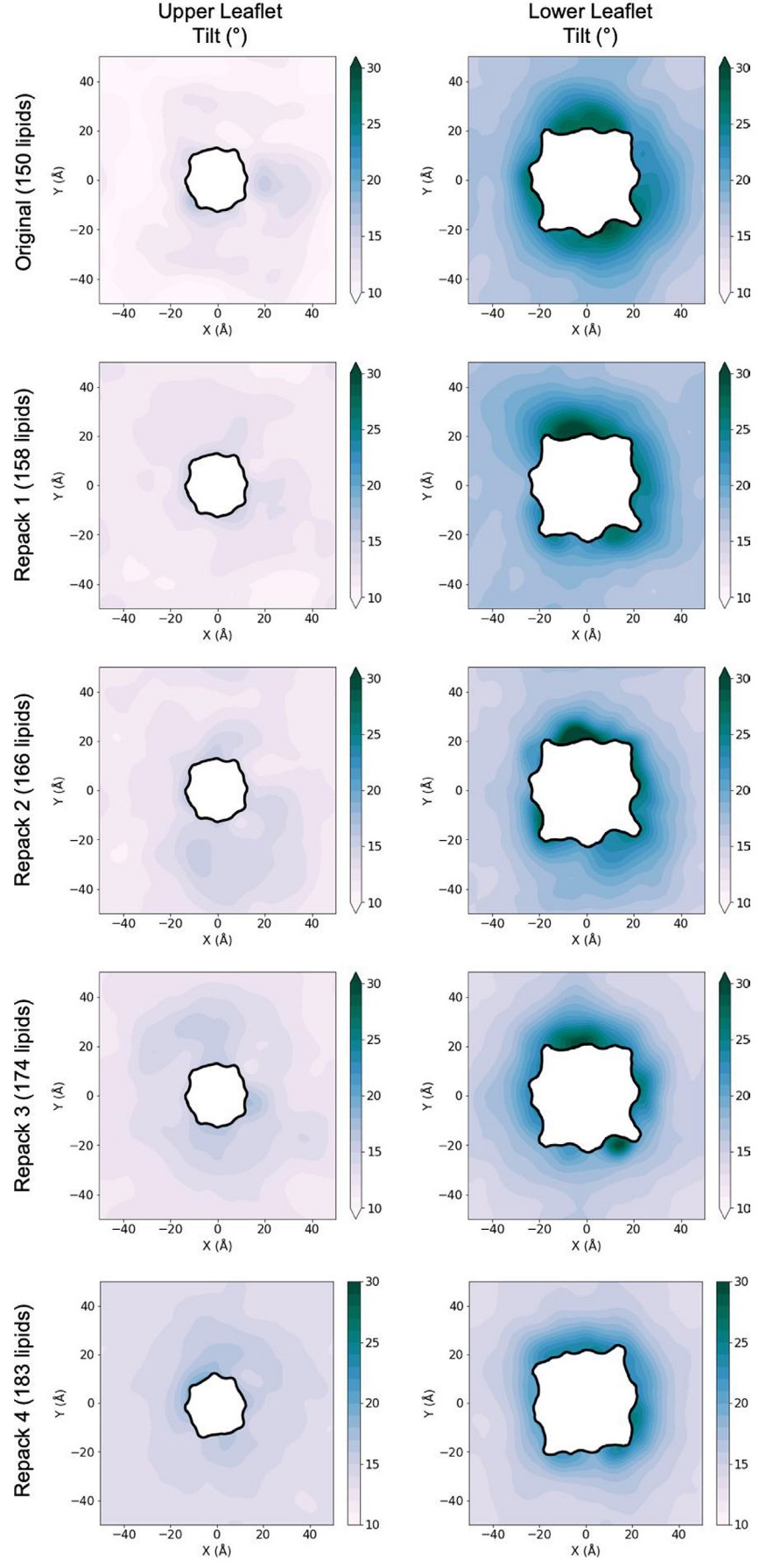

**Appendix 1—figure 2.** Leaflet tilt surfaces for the original restrained 2L0J simulation and restrained Repack simulations 1–4. The upper leaflet has 200 lipids, and the lower leaflet lipid count is indicated in parentheses.

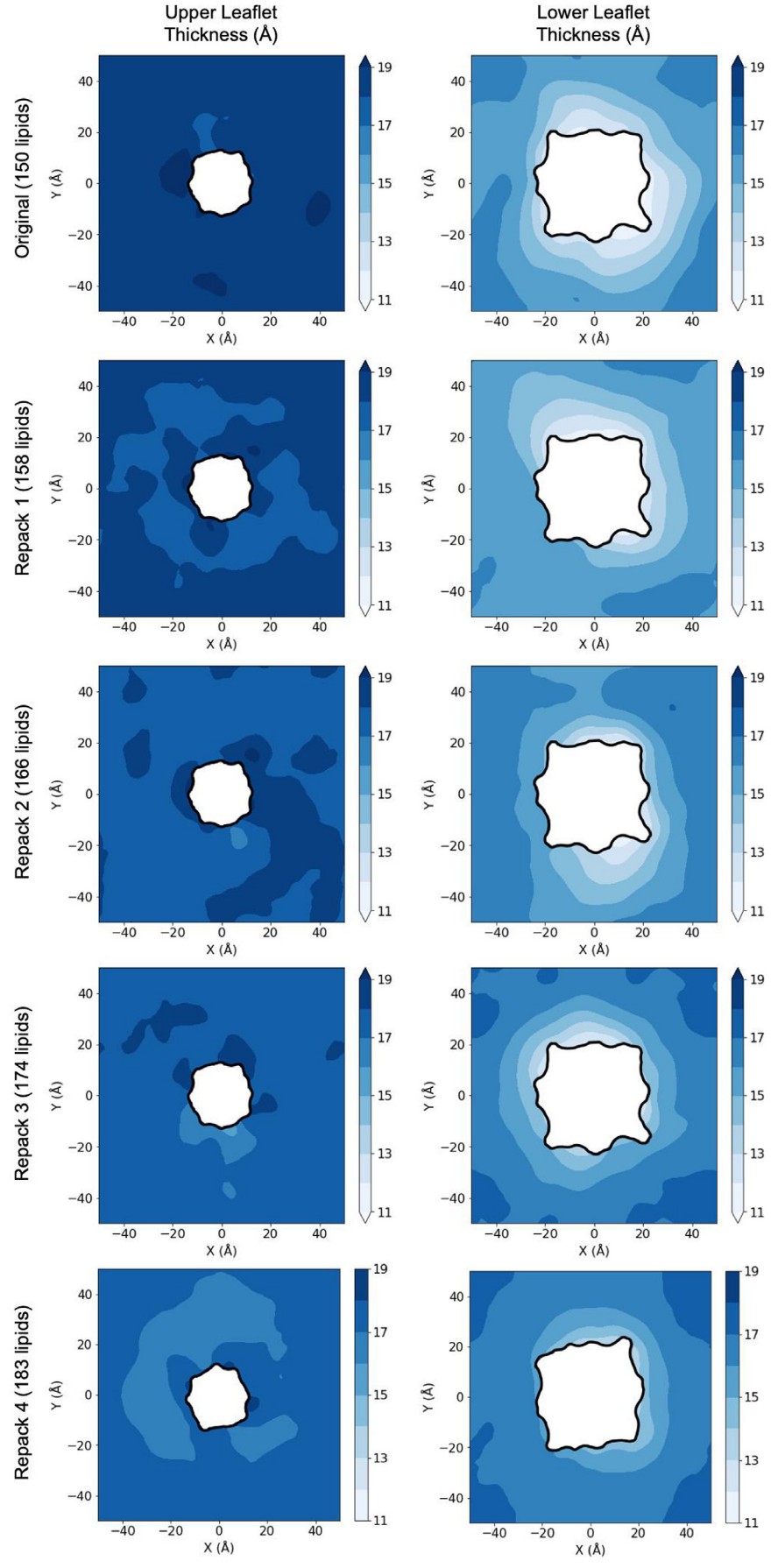

**Appendix 1—figure 3.** Leaflet thicknesses for the original restrained 2L0J simulation and restrained Repack simulations 1–4. The upper leaflet has 200 lipids, and the lower leaflet lipid count is indicated in parentheses.

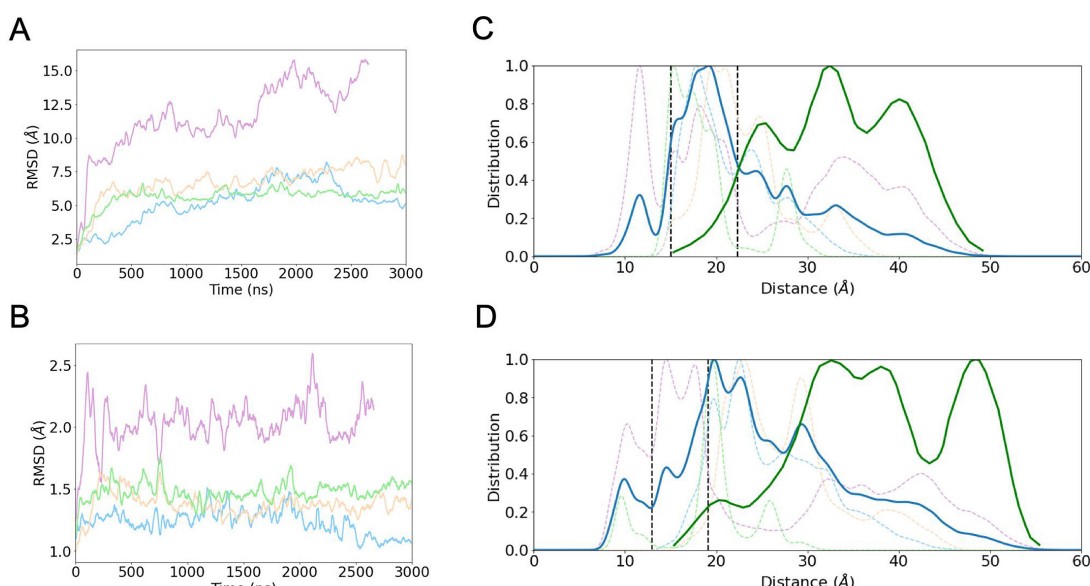

**Appendix 1—figure 4.** Amphipathic helix (AH) domain mobility in Repack 4, unrestrained 2L0J simulations. (**A**) Trajectory RMSDs for the AH domain. Independent replicates are color coded. (**B**) Trajectory RMSDs for the transmembrane (TM) domain. (**C**) I51–I51 distance distributions from unrestrained simulations. Dashed curves are color coded per trajectory. The aggregate distance distribution across all replicates is solid blue. DEER data from the Howard lab in solid green (see MT **Figure 2B** for source data). Vertical dashed lines represent the adjacent and diagonal distances for I51 in 2L0J. (**D**) F55–F55 distance distributions with the same color coding as panel C.

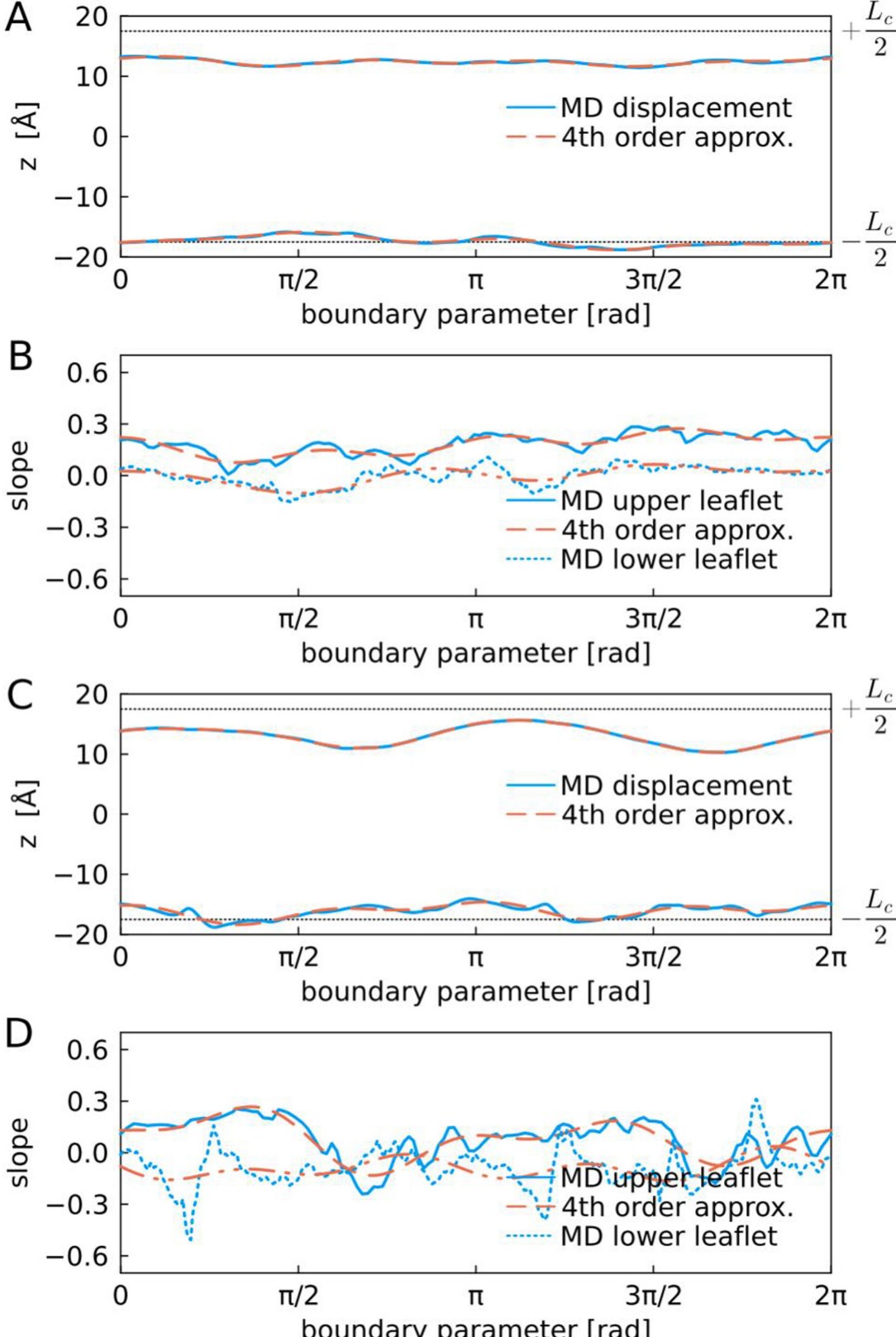

**Appendix 1—figure 5.** Boundary conditions extracted from molecular dynamics (MD) simulations. (**A, B**) Membrane–protein boundary information for the fourfold 2L0J structure. (**C, D**) Membrane–protein boundary information for the twofold 2N70 structure. Upper and lower bounds of the mean hydrophobic membrane core

*Appendix 1—figure 5 continued on next page*

(defined by the surface separating the lipid headgroups from the acyl chain) along the membrane–protein contact curve for 2L0J (**A**) and 2N70 (**C**). Solid blue: mean MD membrane hydrophobic boundaries. Dashed red: fourth-order Fourier series approximation of the MD boundary. Slopes of the mean MD hydrophobic surfaces at the protein in the direction normal to the boundary for 2L0J (**B**) and 2N70 (**D**). Blue solid (dotted): upper (lower) leaflet slopes. Dashed red: fourth-order Fourier series approximation of the MD slopes. The boundary parameter is the length around the boundary, normalized to $2\pi$.

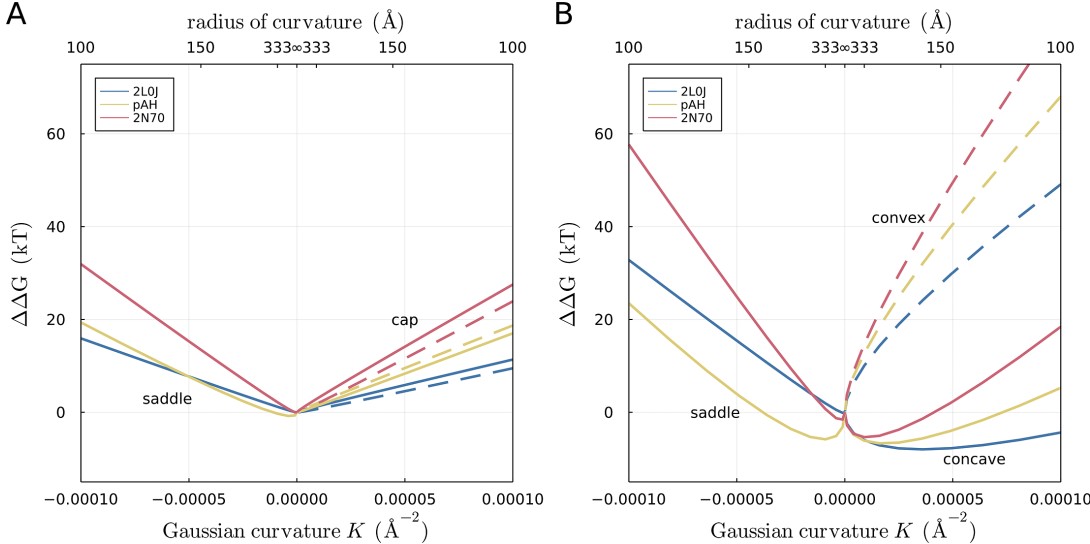

**Appendix 1—figure 6.** Influence of cholesterol enrichment on membrane curvature sensing by different M2 channel conformations. Membrane parameters from **Table 2** were used to calculate the membrane bending energy for moving from a flat membrane region to regions of differing curvature. (**A**) A 0% cholesterol membrane with 28.5 Å core thickness. (**B**) A 50% cholesterol membrane with a 35-Å core thickness. The 30% cholesterol case is shown in **Figure 11A**. Positive Gaussian curvatures $K$ correspond to concave (solid curves) or convex (dashed curves) spherical caps, while negative Gaussian curvature values correspond to saddles.

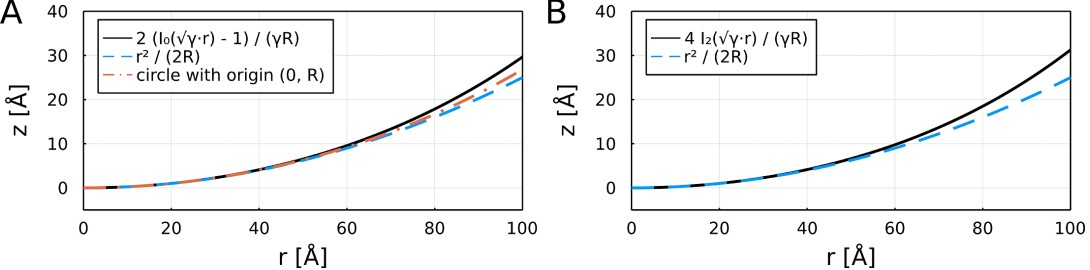

**Appendix 1—figure 7.** Membrane shapes which approximate an ideal spherical cap or saddle. (**A**) Dash–dotted red: circle of radius $R = 200$ Å. Dashed blue: quadratic approximation of circle, valid for $r \ll R$. Solid black: the $m = 0$ analytic biharmonic solution finite at the origin behaves quadratically for $\sqrt{\gamma}r \lesssim 1$. (**B**) Dashed blue: the ideal saddle $z = \cos(2\phi)\, r^2/(2R)$ is quadratic along the $X$ and $Y$ axes with radii of curvature $R$ and $-R$, respectively. Solid black: the $m = 2$ analytic biharmonic solution diverges from quadratic when $r \gtrsim \sqrt{1/\gamma} \approx 60.2$ Å. The numeric solutions from the membrane solver are not shown because they are indistinguishable from the analytic solutions at this scale.

