## [Editor Report]

In this important study, the authors combined atomistic simulations and continuum mechanics models to probe how structural features of the M2 channel impact the local membrane properties and stability of the channel in membranes of different curvatures. The level of evidence provided by the computational analysis is convincing, and the insights can potentially lead to novel strategies that screen for drug molecules that stabilize fission-incompetent conformations of the M2 channel. The multi-scale approach will find utility to many problems in membrane reshaping.

---

## [Decision Letter]

**Decision letter after peer review:**

Thank you for submitting your article "Membrane curvature sensing and symmetry breaking of the M2 proton channel from Influenza A" for consideration by *eLife*. Your article has been reviewed by 3 peer reviewers, one of whom is a member of our Board of Reviewing Editors, and the evaluation has been overseen by Kenton Swartz as the Senior Editor. The following individuals involved in review of your submission have agreed to reveal their identity: Alexander Sodt (Reviewer #2); Olaf S Andersen (Reviewer #3).

Essential revisions (for the authors):

1) Further clarifications regarding the choice of some of the simulation details, such as the lipid composition in atomistic simulations, impact of symmetry constraint in the protein structure in some simulations, extraction of boundary condition for the continuum calculations, and the definition of lipid tilt.

2) The authors should check if thinning, and the boundary, is sensitive to inter-leaflet-lipid imbalance.

*Reviewer #1 (Recommendations for the authors):*

Both atomistic and continuum mechanics calculations have been conducted with a great level of expertise. The only question I have concerns the materials properties used in the continuum mechanical analysis. Are these based on membranes of similar composition, especially the high cholesterol concentration? Also, several previous studies highlighted the impact of the protein on nearby membrane mechanical properties, such as bending moduli. Is including such spatial dependence of materials properties expected to make any qualitative impact on the conclusion, especially considering the relatively modest energetic preference to membranes of different (Gaussian) curvature?

*Reviewer #2 (Recommendations for the authors):*

Suggestions:

Apart from running a simulation with varied lipid numbers in the upper leaflet, can the continuum model address the problem of leaflet differential stress (see, e.g., Hossein and Deserno 2020)? Perhaps the continuum model can determine the proper change in leaflet-count of the expanded leaflet to minimize the free energy?

The authors should be able to estimate the reduction in NGC preference with a simple two state model of the protein conformations fluctuations around 2-fold symmetry, and the variation of coupling to NGC with symmetry breaking. The unrestrained simulations provide a simple model of the fluctuations around 2-fold symmetry and alternative boundary conditions for the continuum model.

*Reviewer #3 (Recommendations for the authors):*

1. Somewhere in the introduction the authors should refer to Phillips et al. (Nature, 2009), who considered non-planar surfaces.

2. P. 5: It is not clear from the text why there will be a line tension at the domain boundary between nascent bud and the host membrane. Please add a reference, or briefly explain.

3. The authors use the terms concave and convex throughout the manuscript, but do not really specify what they mean (a convex surface would be concave if viewed from the other side). Though a minor point, the last paragraph on P. 20 would benefit from this clarification (and a reference to Figure 1).

4. Throughout the manuscript the authors refer to the M2 channel, which is correct, but none of the authors' conclusions depend on the M2 protein having a catalytic function; it is the structural/dynamic features that are important. So, why not just use the term "M2 protein"?

5. A problem Figures 3 and 9 is that the viewing angles of different membrane deformation profiles are different, e.g., the left-most surface in Figure 3B, as compared with the other two. Given the authors' conclusion that the lower/inner interface is nearly flat, I think all panels should be oriented to emphasize this feature.

The authors should make sure that all symbols used in the figures are defined, e.g., "e" and "c" in Figure 1.

6. P. 9-11: Figures 3 and 4 belong together, and it may be helpful for the reader to refer also to Figure 4 when Figure 3 is introduced. Also, please specify that you plot the hydrophobic thickness, that is not clear unless one has read the Methods section.

7. P. 14, first sentence of second paragraph: the authors need refer to both Figures 5 and 6. Also, isn't 12 degrees the same as 10 degrees; what is the range (standard deviation)

8. P. 19, last sentence: Given the difficulties of calculating the deformation energies, even from molecular dynamics simulations (Fiorin et al., 2020), the authors may wish to cite Sun et al. (Biophys J, 2019) who obtained near-quantitative agreement between the simulation-derived energies and experimental estimates.

9. P.20, line 4: what is meant by "relaxes"? I believe the correct is "adapts".

10. P. 23, third paragraph: the statement "inhibits NGC binding" needs to be rephrased.

---

## [Author Response]

Essential revisions (for the authors):Reviewer #1 (Recommendations for the authors):Both atomistic and continuum mechanics calculations have been conducted with a great level of expertise. The only question I have concerns the materials properties used in the continuum mechanical analysis. Are these based on membranes of similar composition, especially the high cholesterol concentration?

In short, we did not use the most appropriate data in the literature for our continuum calculations, and we have updated our values and given better motivation for our parameters in the revised text.

In the original submission, the original parameters were based on an equilibrium hydrophobic thickness (L_C_) from our current simulations, an experimentally determined compressibility (K_a_) for a pure 100% POPC bilayer (Henriksen et al., 2006), the Evan’s Brush Bottle model to use L_C_ and K_a_ to estimate the bending modulus (K_C_) (Rawicz et al., 2000), and then a simulation-derived relationship to obtain the Gaussian bending modulus (K_G_) from the mean bending modulus (Hu et al., 2012). The 30% cholesterol content in our simulations caused an increase in the thickness and subsequently the bending moduli compared to a pure POPC bilayer. However, the Brush Bottle model may not be ideal for cholesterol rich POPC regarding bending moduli, and we ignored important increases in K_a_ that come from greater sterol content.

In the revised submission, we have modified the continuum parameters to better estimate the true properties of the mixed membrane, and we have provided estimates for 3 different modeled compositions (see Table 2 in revised manuscript). The 30 % composition is explored extensively in the main text, and the 0 % and 50 % are explored in supplemental figures. Membrane parameter values were estimated based on cholesterol doped POPC bilayers, ignoring the small additional amounts of POPG used in our simulations. In Table 2, where a reference is cited the value was taken directly from the paper. In other cases, we had to extrapolate from existing data as described next. We mention that the Sodt paper (Hossein and Sodt, 2023) could be used to directly extract these parameters from our multicomponent simulations.

“Membrane Thickness (L_C_). Ferreira and co-workers provide NMR-derived tail lengths for POPC with increasing amounts of cholesterol, getting 14.5 Å at 34 % cholesterol and 15.3 Å at 50 % cholesterol (Ferreira et al., 2013). Treating the 5.5 % difference going from the lower to higher cholesterol condition and adding that to our MD value of 35 Å bilayer thickness at 30 % cholesterol, our 50 % cholesterol estimate is 37 Å.

Mean bending modulus (K_C_). The K_C_ value for a pure POPC membrane was obtained from averaging five reported values obtained using four different methods: all-atom MD (Drabik et al., 2020; Venable et al., 2015), small-angle X-ray (Kučerka et al., 2006), micropipette aspiration (Henriksen et al., 2006), and flicker noise spectroscopy analysis (Drabik et al., 2020). We estimated the mean bending modulus at 30% cholesterol using the scaling from 0 to 30 % cholesterol from Henriksen, which is 225 % (Henriksen et al., 2006), along with the mean value at 0 % reported in the table. For cholesterol content above 30 %, Pan and co-workers showed that the bending modulus is rather constant for a single chain mono-unsaturated lipid SOPC, where they reported a 4 % increase from 30 to 50 % (Pan et al., 2009), which we use here.

Gaussian bending modulus (K_G_). We use the phenomenological scaling relation between K_C_ and K_G_ reported by Hu and Deserno (K_G_ = -0.9 K_C_) to arrive at K_G_ from the values reported for K_C_. This model is based on a coarse-grained Cooke model of the lipids, and we note that other all-atom approaches to determine this relationship arrive at smaller ratios (Venable et al., 2015). Nonetheless, while we include this parameter in our table for completeness, it actually has no impact on the G values reported here because the total Gaussian curvature is constant in these cases.

Area compression modulus (K_a_). We assume that changes in K_a_ above 30 % cholesterol plateau as observed for L_C_ and K_C_, and we estimate only a 5 % increase from 30 to 50 % cholesterol as reported in Table 2.”

Also, several previous studies highlighted the impact of the protein on nearby membrane mechanical properties, such as bending moduli. Is including such spatial dependence of materials properties expected to make any qualitative impact on the conclusion, especially considering the relatively modest energetic preference to membranes of different (Gaussian) curvature?

Previously, we had to use shell hardening near particular proteins to obtain a quantitative match between continuum membrane models and our all-atom simulations (Argudo et al., 2017); however, this was not the case in the current simulations nor was it the case in our analysis of membrane-deforming TMEM16 proteins (Bethel and Grabe, 2016). It is not obvious to us why some systems require this change and others do not, but the excellent agreement between the updated data presented in this resubmission suggests to us that ignoring spatial dependence in moduli is a reasonable assumption. That said, if the membrane does harden the curvature sensing that we show would be more pronounced.

Reviewer #2 (Recommendations for the authors):Suggestions:Apart from running a simulation with varied lipid numbers in the upper leaflet, can the continuum model address the problem of leaflet differential stress (see, e.g., Hossein and Deserno 2020)? Perhaps the continuum model can determine the proper change in leaflet-count of the expanded leaflet to minimize the free energy?

We thank Reviewer #2 for this insight, as it led us to evaluate the leaflet tensions in our restrained 2L0J simulation. We found there was an imbalance in the leaflet packing, which we addressed with an extensive set of new simulations and new analysis aimed at generating balanced leaflets.

We discuss these findings in the new Results section “Protein footprint asymmetry can lead to differential leaflet stresses” and accompanying appendix. Many of the bilayer features in the repacked simulations are consistent with our original submission, but not all. For instance, while we continue to see large tilt immediately around the amphipathic helices in the lower leaflet and little in the upper leaflet, tilts in both leaflets decay to similar values at the box edge (Appendix – figure 2). The degree of membrane pinch along the membrane-protein contact boundaries are less sensitive to the leaflet packing, as demonstrated by the surface heights (Appendix – figure 1).

Determining the proper change in leaflet count is quite difficult. We are actively extending our continuum model to address questions of differential leaflet strain and coupled lipid tilt, which may allow us to estimate changes in leaflet-count, but this is a significant undertaking beyond the scope of this resubmission.

The authors should be able to estimate the reduction in NGC preference with a simple two state model of the protein conformations fluctuations around 2-fold symmetry, and the variation of coupling to NGC with symmetry breaking. The unrestrained simulations provide a simple model of the fluctuations around 2-fold symmetry and alternative boundary conditions for the continuum model.

As discussed in the last reply, we cannot do this without an estimate of the energy difference between the 4-fold and 2-fold states, and it is beyond our current scope.

Reviewer #3 (Recommendations for the authors):1. Somewhere in the introduction the authors should refer to Phillips et al. (Nature, 2009), who considered non-planar surfaces.

We have now referenced the Phillips review in the introduction.

2. P. 5: It is not clear from the text why there will be a line tension at the domain boundary between nascent bud and the host membrane. Please add a reference, or briefly explain.

The domain boundary is characterized by regions with different lipid compositions which gives rise to a chemical potential for lipids moving from one phase to the other. This phase boundary is unfavorable and the system moves to minimize it, which can occur quickly as highlighted in work on liposomes (Veatch and Keller, 2003). The role of line tension arising from phase separation has featured prominently in general models of fission/fusion (Ursell et al., 2009), endosomal budding (Liu et al., 2006), and viral budding (Tzlil et al., 2004). The magnitude of this tension can vary, however, depending on the particular system.

3. The authors use the terms concave and convex throughout the manuscript, but do not really specify what they mean (a convex surface would be concave if viewed from the other side). Though a minor point, the last paragraph on P. 20 would benefit from this clarification (and a reference to Figure 1).

We have defined our convention in Figure 1, and throughout the entire manuscript.

4. Throughout the manuscript the authors refer to the M2 channel, which is correct, but none of the authors' conclusions depend on the M2 protein having a catalytic function; it is the structural/dynamic features that are important. So, why not just use the term "M2 protein"?

We have added a note on our terminology.

5. A problem Figures 3 and 9 is that the viewing angles of different membrane deformation profiles are different, e.g., the left-most surface in Figure 3B, as compared with the other two. Given the authors' conclusion that the lower/inner interface is nearly flat, I think all panels should be oriented to emphasize this feature.

We have changed the orientation of all three proteins in Figure 4B (originally Figure 3B). The viewing angle was also tilted to reveal more of the lower leaflet so that the flatness can be better appreciated. The orientations in Figures 4 and 11 (originally Figure 9) are not the same, as we want to highlight the deformation profiles in curved surfaces, and that requires drawing different orientations.

The authors should make sure that all symbols used in the figures are defined, e.g., "e" and "c" in Figure 1.

We have defined all symbols in the figures.

6. P. 9-11: Figures 3 and 4 belong together, and it may be helpful for the reader to refer also to Figure 4 when Figure 3 is introduced. Also, please specify that you plot the hydrophobic thickness, that is not clear unless one has read the Methods section.

This comment refers to new Figures 4 and 5. We have kept the figures separate to keep the amount of information in each manageable; but we now reference them both simultaneously when we discuss the 3D surfaces in Figure 4. Additionally, we specifically indicate that the thickness plots are hydrophobic thickness.

7. P. 14, first sentence of second paragraph: the authors need refer to both Figures 5 and 6. Also, isn't 12 degrees the same as 10 degrees; what is the range (standard deviation)

This entire section has been substantially rewritten due to the new calculations. We now plot the radial profile for the surface deflection heights and lipid tilt values together (new Figure 8). We have chosen not to compute the standard deviation of the tilt distributions, which we expect to be large, but the mean of the distributions is very tightly clustered in some regions, especially far from the protein where it is less than 1 degree (see Figure 8B, 2LOJ upper leaflet). We now refer to minimum mean tilt differences of 3° rather than 2°, and we believe that the tight clustering of the data suggests that these values are meaningfully different.

8. P. 19, last sentence: Given the difficulties of calculating the deformation energies, even from molecular dynamics simulations (Fiorin et al., 2020), the authors may wish to cite Sun et al. (Biophys J, 2019) who obtained near-quantitative agreement between the simulation-derived energies and experimental estimates.

We have now added the citation to the manuscript by Sun and co-workers (Sun et al., 2019), as this is an important step that in building accurate and predict models of membrane deformation by comparing directly to experiment, not just theory to simulation.

9. P.20, line 4: what is meant by "relaxes"? I believe the correct is "adapts".

We agree, and this word has been changed.

10. P. 23, third paragraph: the statement "inhibits NGC binding" needs to be rephrased.

We changed this phrase to “inhibit M2 localization to NGC”.

References

Argudo, D., Bethel, N. P., Marcoline, F. V., Wolgemuth, C. W., and Grabe, M. (2017). New Continuum Approaches for Determining Protein-Induced Membrane Deformations. Biophys J, 112(10), 2159-2172. https://doi.org/10.1016/j.bpj.2017.03.040

Bethel, N. P., and Grabe, M. (2016). Atomistic insight into lipid translocation by a TMEM16 scramblase. Proc Natl Acad Sci U S A, 113(49), 14049-14054. https://doi.org/10.1073/pnas.1607574113

Drabik, D., Chodaczek, G., Kraszewski, S., and Langner, M. (2020). Mechanical Properties Determination of DMPC, DPPC, DSPC, and HSPC Solid-Ordered Bilayers. Langmuir, 36(14), 3826-3835. https://doi.org/10.1021/acs.langmuir.0c00475

Ferreira, T. M., Coreta-Gomes, F., Ollila, O. H., Moreno, M. J., Vaz, W. L., and Topgaard, D. (2013). Cholesterol and POPC segmental order parameters in lipid membranes: solid state 1H-13C NMR and MD simulation studies. Phys Chem Chem Phys, 15(6), 19761989. https://doi.org/10.1039/c2cp42738a

Gerl, M. J., Sampaio, J. L., Urban, S., Kalvodova, L., Verbavatz, J. M., Binnington, B., Lindemann, D., Lingwood, C. A., Shevchenko, A., Schroeder, C., and Simons, K. (2012). Quantitative analysis of the lipidomes of the influenza virus envelope and MDCK cell apical membrane. J Cell Biol, 196(2), 213-221. https://doi.org/10.1083/jcb.201108175

Henriksen, J., Rowat, A. C., Brief, E., Hsueh, Y. W., Thewalt, J. L., Zuckermann, M. J., and Ipsen, J. H. (2006). Universal behavior of membranes with sterols. Biophys J, 90(5), 1639-1649. https://doi.org/10.1529/biophysj.105.067652

Hossein, A., and Sodt, A. J. (2023). Membraneanalysis. jl: A Julia package for analyzing molecular dynamics simulations of lipid membranes. Journal of Open Source Software, 8(87), 5380.

Hu, M., Briguglio, J. J., and Deserno, M. (2012). Determining the Gaussian curvature modulus of lipid membranes in simulations. Biophys J, 102(6), 1403-1410. https://doi.org/10.1016/j.bpj.2012.02.013

Ivanova, P. T., Myers, D. S., Milne, S. B., McClaren, J. L., Thomas, P. G., and Brown, H. A. (2015). Lipid composition of viral envelope of three strains of influenza virus – not all viruses are created equal. ACS Infect Dis, 1(9), 399-452. https://doi.org/10.1021/acsinfecdis.5b00040

Kim, S. S., Upshur, M. A., Saotome, K., Sahu, I. D., McCarrick, R. M., Feix, J. B., Lorigan, G. A., and Howard, K. P. (2015). Cholesterol-Dependent Conformational Exchange of the CTerminal Domain of the Influenza A M2 Protein. Biochemistry, 54(49), 7157-7167. https://doi.org/10.1021/acs.biochem.5b01065

Kučerka, N., Tristram-Nagle, S., and Nagle, J. F. (2006). Structure of fully hydrated fluid phase lipid bilayers with monounsaturated chains. J Membr Biol, 208(3), 193-202.

Latorraca, N. R., Callenberg, K. M., Boyle, J. P., and Grabe, M. (2014). Continuum approaches to understanding ion and peptide interactions with the membrane. J Membr Biol, 247(5), 395-408. https://doi.org/10.1007/s00232-014-9646-z

Liu, J., Kaksonen, M., Drubin, D. G., and Oster, G. (2006). Endocytic vesicle scission by lipid phase boundary forces. Proc Natl Acad Sci U S A, 103(27), 10277-10282. https://doi.org/10.1073/pnas.0601045103

Pan, J., Tristram-Nagle, S., and Nagle, J. F. (2009). Effect of cholesterol on structural and mechanical properties of membranes depends on lipid chain saturation. Phys Rev E Stat Nonlin Soft Matter Phys, 80(2 Pt 1), 021931. https://doi.org/10.1103/PhysRevE.80.021931

Rawicz, W., Olbrich, K. C., McIntosh, T., Needham, D., and Evans, E. (2000). Effect of chain length and unsaturation on elasticity of lipid bilayers. Biophys J, 79(1), 328-339. https://doi.org/10.1016/S0006-3495(00)76295-3

Sun, D., Peyear, T. A., Bennett, W. F. D., Andersen, O. S., Lightstone, F. C., and Ingolfsson, H. I. (2019). Molecular Mechanism for Gramicidin Dimerization and Dissociation in Bilayers of Different Thickness. Biophys J, 117(10), 1831-1844. https://doi.org/10.1016/j.bpj.2019.09.044

Tzlil, S., Deserno, M., Gelbart, W. M., and Ben-Shaul, A. (2004). A statistical-thermodynamic model of viral budding. Biophys J, 86(4), 2037-2048. https://doi.org/10.1016/S00063495(04)74265-4

Ursell, T. S., Klug, W. S., and Phillips, R. (2009). Morphology and interaction between lipid domains. Proc Natl Acad Sci U S A, 106(32), 13301-13306. https://doi.org/10.1073/pnas.0903825106

Veatch, S. L., and Keller, S. L. (2003). Separation of liquid phases in giant vesicles of ternary mixtures of phospholipids and cholesterol. Biophys J, 85(5), 3074-3083. https://doi.org/10.1016/S0006-3495(03)74726-2

Venable, R. M., Brown, F. L. H., and Pastor, R. W. (2015). Mechanical properties of lipid bilayers from molecular dynamics simulation. Chem Phys Lipids, 192, 60-74. https://doi.org/10.1016/j.chemphyslip.2015.07.014